# A China dataset of soil properties for land surface modeling (version 2, CSDLv2)

Gaosong Shi[1], Wenye Sun[1], Wei Shangguan[1,*], Zhongwang Wei[1], Hua Yuan[1], Lu Li[1], Xiaolin Sun[2], Ye Zhang[1], Hongbin Liang[1], Danxi Li[1], Feini Huang[1], Qingliang Li[1,3], and Yongjiu Dai[1]

[1]Southern Marine Science and Engineering Guangdong Laboratory (Zhuhai), Guangdong Province Key Laboratory for Climate Change and Natural Disaster Studies, School of Atmospheric Sciences, Sun Yat-sen University, Guangzhou 510275, China;
[2]School of Geography and Planning, Sun Yat-sen University, Guangzhou 510275, China;
[3]College of Computer Science and Technology, Changchun Normal University, Changchun 130032, China

*Correspondence to*: Wei Shangguan (Email: shgwei@mail.sysu.edu.cn)

**Abstract.** Accurate and high-resolution spatial soil information is crucial for efficient and sustainable land use, management, and conservation. Since the establishment of digital soil mapping (DSM) and the GlobalSoilMap working group, significant advances have been made in spatial soil information globally. However, accurately predicting soil variation over large and complex areas with limited samples remains a challenge, especially for China, which has diverse soil landscapes. To address this challenge, we utilized 11 209 representative multi-source legacy soil profiles (including the Second National Soil Survey of China, World Soil Information Service, First National Soil Survey of China, and regional databases) and high-resolution soil-forming environment characterization. Using advanced ensemble machine learning and a high-performance parallel computing strategy, we developed comprehensive maps of 23 soil physical and chemical properties at six standard depth layers from 0 to 2 meters in China with a 90 m spatial resolution (China dataset of soil properties for land surface modeling version 2, CSDLv2). Data-splitting and independent samples validation strategies were employed to evaluate the accuracy of the predicted maps quality. The results showed that the predicted maps were significantly more accurate and detailed compared to traditional soil type linkage methods (i.e., CSDLv1, the first version of the dataset), SoilGrids 2.0, and HWSD 2.0 products, effectively representing the spatial variation of soil properties across China. The prediction accuracy of soil properties all depth interval ranged from good to moderate, with Model Efficiency Coefficients for most soil properties median ranging from 0.29 to 0.70 during data-splitting validation and from 0.25 to 0.84 during independent sample validation. The wide range between the 5% lower and 95% upper prediction limits may indicate substantial room for improvement in current predictions. The relative importance of environmental covariates in predictions varied with soil properties and depth, indicating the complexity of interactions among multiple factors in the soil formation processes. As the soil profiles used in this study mainly originate from the Second National Soil Survey of China during 1970s and 1980s, they could provide new perspectives of soil changes together with existing maps based on 2010s soil profiles. The findings make important contributions to the GlobalSoilMap project and can also be used for regional Earth system modeling and land surface modeling to better represent the role of soil in hydrological and biogeochemical cycles in China. This dataset is freely available at https://www.scidb.cn/s/ZZJzAz or https://doi.org/10.11888/Terre.tpdc.301235 (Shi et al, 2024).

## 1 Introduction

Soil plays a pivotal role in earth's systems, facilitating the cycling of water, energy, and carbon across varying temporal and spatial scales. Its significance lies in regulating ecosystems by providing vital nutrients to living organisms, storing and cycling water, heat, carbon, and essential nutrients, and serving as a medium for vegetation growth and structural support (Chaney et al., 2019; Crow et al., 2012). Soil data are essential for land surface models (LSMs), which form a part of Earth system models (ESMs) (Dai et al., 2019b; Luo et al., 2016). The diverse range of soil properties and their precise representation are crucial for robust land surface modeling, influencing various environmental, agricultural, and ecological assessments. There is an urgent need for detailed, accurate, and up-to-date soil information to develop solutions for these challenges and to inform decision-making related to natural resource management (Arrouays et al., 2014; Dai et al., 2019b; Li et al., 2024).

In recent years, the national and global maps of soil properties have gained significant traction in research (Arrouays et al., 2017), with a surge of studies focusing on mapping one or more soil properties at high resolutions such as 90 meters spanning various countries. These include large-scale endeavors in Australia (Grundy et al., 2015; Viscarra Rossel et al., 2015), France (Chen et al., 2023; Mulder et al., 2016), Chile (Dinamarca et al., 2023; Padarian et al., 2017), Japan (Yamashita et al., 2024), Netherlands (Helfenstein et al., 2024) and the United States (Ramcharan et al., 2018; Thompson et al., 2020). Chaney et al., (2019) developed 30 m probabilistic maps of soil properties across the United States. Denmark has also developed national maps of soil texture at a finer 30 m resolution (Adhikari et al., 2013). Additionally, broader-scale resolution maps, ranging from 250 to 5000 m, have also been investigated at the national level, exemplified by Brazil (Gomes et al., 2019). These efforts have been expanded to continental scales, including Africa (Hengl et al., 2015, 2021) and Europe (Heuvelink et al., 2016), and ultimately to global levels, as seen in datasets such as the Global Soil Dataset for use in Earth system models (GSDE, Shangguan et al., 2014), the Harmonized World Soil Database version 2.0 (HWSD 2.0, FAO & IIASA, 2023), and SoilGrids 2.0 (Poggio et al., 2021).

Shangguan et al., (2013) pioneered the development of a comprehensive soil characteristics dataset specifically designed for land surface modeling over China (i.e., China Soil Dataset for Land Surface Modelling, CSDLv1, the first version dataset of this study). This dataset, based on 8 979 legacy soil profiles and the soil map of China (1:1,000,000), employs the conventional polygon linkage method (Batjes, 1995, 2002; Shangguan et al., 2012) to develop soil physical and chemical properties. It provides a spatial resolution of 30 arc-second (about 1 km at equator) and includes over 20 properties at 8 vertical soil depths (Shangguan et al., 2013). The dataset has been successfully applied in various fields. Despite its significant contributions to regional land surface modeling and geoscientific research, over time, several issues and shortcomings have been identified. First, while the dataset utilized soil profiles solely from the Second National Soil Survey of China (1979-1985), there is now a broader array of available soil profiles, including those from the World Soil Information Service (WoSIS, (Batjes et al., 2020)), regional database (Shangguan et al., 2012) and the First National Soil Survey of China (National Soil Survey Office, 1964). The integration of these soil profiles promises to substantially enhance the spatial representation and coverage of the dataset. Second, this dataset relies on the traditional polygon linkage method based on soil transformation rules (Shangguan et al., 2013, 2014), where results heavily depend on the accuracy of soil classification maps and are estimated as the average of a soil class or polygon, leading to discontinuous spatial estimates. The emergence of digital soil mapping (DSM) techniques (McBratney et al., 2003), particularly the success of machine learning in large-

scale spatial prediction (Hengl et al., 2017; Poggio et al., 2021; Yan et al., 2020), presents a methodological advancement for this study. Recent studies indicate that advanced machine learning models often outperform simpler ones, with the size of the sample also emerging as a crucial factor influencing model performance (Padarian et al., 2020).

For China, mapping datasets encompassing one or multiple soil properties have already been developed. Liang et al., (2019) and Chen et al., (2019) both developed high-resolution grid maps across China based on about 5,000 legacy soil profiles collected from the Second National Soil Survey of China, providing more detailed information for areas with spatial heterogeneity. However, Liang et al. (2019) focused solely on spatial estimates for soil organic carbon in the topsoil (0-20 cm layer), while Chen et al. (2019) concentrated solely on spatial estimates for soil pH in the same layer. Both studies lack estimations for other soil property variables and deeper soil layers. Approximately 4 000 legacy soil profiles were utilized by Zhou et al., (2019a) to develop a high-resolution national-scale dataset for total nitrogen in the topsoil (0-20 cm layer) with a 90 m resolution using machine learning methods. Similarly, Song et al., (2020) used over 5 000 soil profiles from the 2010s to produce high-resolution maps of soil organic carbon at six standard depths (0-5, 5-15, 15-30, 30-60, 60-100, and 100-200 cm) across China, achieving explained variances ranging from 0.16 to 0.57. Besides, Liu et al., (2022a) also employed machine learning methods to develop China's inaugural high-resolution national soil information grid dataset at a 90 m resolution, utilizing soil samples from the most recent national soil series survey (2009-2019). This dataset has significantly contributed to soil management, agricultural production, hydrological modeling, ecological development, and climate change mitigation. However, the study relied solely on a constrained set of about 4 500 soil profiles collected during the recent national soil survey, generating national grid maps for only some fundamental soil properties, including pH ($H_2O$), organic carbon, cation exchange capacity, total nitrogen, total phosphorus, total potassium, bulk density, gravel content, soil texture, and soil thickness. The limitations stem from the absence of more comprehensive national grid maps for soil properties, including the fractions of total phosphorus and potassium readily available for plant absorption (Available phosphorus, AP; Available potassium, AK), an index of the potential capacity of the soil to supply nitrogen (Alkali-hydrolysable nitrogen, AN), porosity, and others, imposing constraints on applications that necessitate a broader spectrum of soil properties information. Additionally, there are abundant legacy soil profiles stored in global or regional databases (e.g., WoSIS, (Batjes et al., 2020)). These legacy soil profiles serve as a primary data source for digital soil mapping (Lagacherie et al., 2024; Song et al., 2020; Yang et al., 2022). For China, the Second National Soil Survey serves as a significant source of legacy soil profiles, offering valuable insights into soil properties and characteristics (Shangguan et al., 2013). Therefore, these rich legacy soil profile data should be fully utilized, as they better reflect historical mapping results, providing a new perspective for studying temporal changes in soil properties (Song et al., 2020). In summary, the existing dataset has several limitations, including its reliance on the traditional polygon linkage method, a limited number of soil profile samples, and the fact that it only contains basic soil property variables, lacking more comprehensive soil properties. Given these limitations, there is a compelling need to develop a new version of the dataset to address these challenges.

This paper aims to develop a new version of CSDL (CSDLv2), with comprehensive soil physical and chemical properties for China at a 90 m resolution. This work builds on its previous version (CSDLv1, Shangguan et al., 2013), integrating advanced machine learning algorithms, multi-source soil profile samples, and high-resolution environmental covariates related to soil formation. The key advancements of this second edition dataset, compared to the first edition, are as follows:

1. Integration of multi-source soil profile samples, including data from the Second National Soil Survey of China (Shangguan et al., 2013), the World Soil Information Service (Batjes et al., 2020), the First National Soil Survey of China (National Soil Survey Office, 1964), and regional databases (Shangguan et al., 2012), enhancing the spatial representation of soil profiles, rather than relying solely on data from the Second National Soil Survey as in CSDLv1.

2. Application of advanced machine learning methods, replacing the conventional soil polygon linkage method used in CSDLv1.

3. Consideration of high-resolution environmental covariates as predictors for the machine learning models, allowing the model to capture more detailed spatial relationships between soil properties and environmental factors.

4. As a result of the improvements in points 1-3, the spatial resolution has been enhanced from the original 1 km to 90 m, providing more detailed and accurate spatial predictions of soil properties.

Additionally, compared to existing datasets, this second edition offers a major innovation: over 20 comprehensive soil property variables were developed, while most current research focuses on mapping only a few basic soil properties.

## 2 Materials and Methods

The workflow of this study is shown in Fig. 1. Five main processes are involved in this framework:

1. Harmonizing and preparing soil point data and environmental covariates.

2. Incorporating laboratory measurements of multiple soil profiles and overlaying them with covariates to generate a regression matrix for modeling.

3. Using cross-validation to obtain optimal modeling parameters.

4. Fitting prediction models based on the regression matrix.

5. Applying spatial prediction models using high-resolution covariates and evaluating the models using data-splitting and independent sample validation, as well as uncertainty maps.

### 2.1 Study area and soil profiles

### 2.1.1 Study area

China, located in East Asia along the west coast of the Pacific Ocean, extends from 73°33' to 135°05' E longitude and from 3°51' to 53°33' N latitude, covering an east-west distance of about 5,000 km and featuring a continental coastline exceeding 18,000 km. The terrain of the land area of China exhibits a distinctive "ladder" pattern, with higher elevations in the west descending to lower elevations in the east, as shown in Fig. 2(b). Mountains, plateaus, and hills comprise about 67% of the land area, while basins and plains make up the remaining 33% (Qin et al., 2016). China's topography is highly complex, encompassing an array of landforms such as extensive mountain ranges, vast plateaus, fertile plains, and deep basins. This diverse landscape is further complicated by a range of climatic zones determined by variations in temperature, precipitation, and altitude. These zones include temperate, subtropical, and tropical climates, with the temperate zone being the largest (Fan et al., 2016). Given the complexity and diversity

of China's geographical and climatic conditions, the study of soil properties mapping across this vast nation is of paramount importance.

### 2.1.2 Soil profiles

Typical soil profiles representing main soil-landscapes were collected from four data sources: Second National Soil Survey of China (SNSSC, (National Soil Survey Office, 1996)), World Soil Information Service (WoSIS, (Batjes et al., 2020)), regional datasets (Shangguan et al., 2012), and First National Soil Survey of China (FNSSC, (National Soil Survey Office, 1964)). A total of 11 209 soil profiles were gathered, with distribution details as follows: 8 979 from SNSSC, 1 540 from WoSIS database, 614 from regional datasets, and 76 from FNSSC. Their spatial distribution is illustrated in Fig. 2(a), with different colors representing each data source. The soil property variables considered in this study are listed in Table 1. SNSSC, conducted primarily between 1979 and 1985, provided the majority of soil profiles, although coordinates were approximated due to GPS limitations at the time, impacting mapping accuracy (Lagacherie et al., 2024). Shi et al. (2024) improved the location accuracy of soil profiles in SNSSC by aligning detailed profile descriptions with environmental covariates. WoSIS, managed by the International Soil Reference and Information Centre (ISRIC), is a comprehensive global database that consolidates soil profile data from various sources under a common standard (Batjes et al., 2020). These data are standardized and harmonized to facilitate global soil research and enhance the accuracy of digital soil mapping efforts. It is worth noting that WoSIS contains soil profiles from the SNSSC. The following approach was employed to determine and eliminate potentially duplicate soil profiles in the WoSIS database that may overlap with those in the SNSSC: soil profiles were considered duplicates if they had identical depths of soil horizons or included at least three identical depths, exhibited similar soil property values, and had close geographic coordinates (latitude and longitude). Consequently, 101 duplicate soil profiles were removed from the WoSIS database, leaving 1 540 soil profiles for this study. The regional dataset was collected from five areas in 2008 and 2009 (Shangguan et al., 2012). FNSSC, initiated in 1958, laid the foundation for China's soil science database and agricultural soil classification. The laboratory methods for soil profile data from the SNSSC and WoSIS databases are detailed in Shangguan et al. (2013) and Batjes et al. (2020), respectively. All data are exclusively from soil profiles, with no inclusion of boreholes or augerings. The regional database includes only surface data, while the SNSSC and WoSIS datasets contain full soil profiles. Because soil profiles are extracted from soil survey books of FNSSC or SNSSC, there may be one or several soil profiles for each soil type. As a result, though there is no sampling design for the major data sources, it may be considered soil type-based stratified sampling for the final soil profile database. For soil properties sensitive to temporal changes, such as soil pH, organic carbon content (OC), cation exchange capacity (CEC), total nitrogen (TN), total phosphorus (TP), total potassium (TK), alkali-hydrolysable nitrogen (AN), available phosphorus (AP), and available potassium (AK), we used only soil profile data from the SNSSC. In contrast, for properties less sensitive to temporal changes, such as sand, silt, clay, bulk density (BD), gravel, and porosity, we combined data from multiple sources. Since most soil profiles are from the SNSSC, the maps in CSDLv2 mainly represent the status of soil in the 1980s. The probability density distribution of topsoil (0-5 cm) properties from different data sources is provided in Fig. S1. To align with international soil mapping standards, a continuous depth function using equal-area splines was

applied to horizon data, defining six standard layers (0-5, 5-15, 15-30, 30-60, 60-100, and 100-200 cm) (Arrouays et al., 2014; Arrouays et al., 2015). Detailed descriptions of the equal-area splines can be found in Bishop et al. (1999) and Liu et al. (2022a).

## 2.2 Environmental covariates

Following the *SCORPAN* (soil, climate, organisms, topography, parent material, age and location) concept (McBratney et al., 2003), over 150 environmental covariates associated with soil formation were collected to investigate the spatial distribution of soil properties for this work. A summary of some high-resolution covariates was provided in Table 2, while the complete list can be found in Table S1. These environmental covariates offer information on the factors related to soil properties.

Relief covariates primarily were derived from the MERIT Digital Elevation Model (DEM) dataset (https://hydro.iis.u-tokyo.ac.jp/~yamadai/MERIT_DEM/), a high-precision global DEM with a resolution of 3 arc-seconds (~90 m at the equator), vertically referenced to the EGM96 geoid and horizontally referenced to the World Geodetic System 1984 (Yamazaki et al., 2019). This dataset serves as an improved spaceborne DEM that significantly reduces the major error components found in other DEMs such as NASA's SRTM3 DEM, and Viewfinder Panoramas DEM (Li et al., 2023). Based on this DEM, other relief covariates such as slope, plan curvature, profile curvature, and terrain wetness index were calculated using SAGA GIS (Conrad et al., 2015).

Organism-related covariates were primarily sourced from six datasets: The Landsat 8 Collection 2 Level 2 (LC08C02), MODIS, GLOBELLAND30, the Global Accessibility Map, and GlobCover. L8C2L2 is an advanced satellite data product released by the United States Geological Survey (USGS). Landsat 8, part of the Landsat satellite series, is specifically designed for Earth observation and monitoring. Collection 2 represents an updated version of Landsat data products, incorporating various improvements and enhancements. High-resolution data such as NDVI, NDWI, Band 5, and Band 7 with 90m spatial resolution were obtained from this database via the Google Earth Engine (GEE) platform. MODIS data offers an efficient method for monitoring biosphere changes and understanding Earth's climate system, available at a spatial resolution of 1 km. GLOBELLAND30, a significant achievement from China's global and local land cover remote sensing mapping and technology research project, provides comprehensive global land surface coverage at a 30 m resolution. The Global Accessibility Map illustrates urban and rural population gradients at a 1 km resolution over the years 2000 to present. Developed by the European Space Agency, the GlobCover dataset provides a global land cover map at a 1 km resolution.

Climate factors were chiefly obtained from the MODIS, WorldClim, and CHELSA-climate datasets (DAAC, 2018; Karger et al., 2020), primarily offering at a 1 km spatial resolution and covering the years 1970-2000. Soil factors, i.e., soil classifications, were mainly derived from the Harmonized World Soil Database also available at a 1 km spatial resolution (Nachtergaele et al., 2012). Parent material factors were represented by the depth to bedrock maps and a lithological map (Yan et al., 2020).

All environmental covariates were reprojected to a unified coordinate reference system, specifically Goode's homolosine projection applied to the World Geodetic System (WGS) 1984 projection. This projection was chosen as it is the most effective at minimizing distortions over land among the equal-area projections available in open-source software (Moreira De Sousa et al., 2019). Additionally, the nearest interpolation and bilinear interpolation algorithms were applied to the subtype data (e.g., vegetation

type) and continuous variables, respectively, to resample these environmental covariates to a raster cell size of 90 m resolution for spatial modeling and map prediction.

    Considering the substantial number of available environmental covariates, those with an absolute Pearson correlation coefficient of less than 0.05 with the target variable were excluded. Subsequently, redundant covariates with a Pearson correlation coefficient greater than 0.8 with any other covariate were removed to eliminate autocorrelation among them. For each pair of

environmental covariates with a correlation exceeding this threshold, only the first one in alphabetical order was retained for the modeling phase (Poggio et al., 2021). This process reduced the initial number of environmental covariates to approximately 80 layers.

    In this study, the Recursive Feature Elimination (RFE) method was implemented using the *sklearn.feature_selection* package in Python, which offers a balanced approach between accuracy and computational efficiency. RFE is a robust technique, widely

recognized for its efficacy in selecting optimal covariate sets for regression tree models (Gomes et al., 2019). The RFE process begins by fitting a model that includes all environmental factors, evaluating its performance, and ranking the covariates based on their importance. The least significant factors are systematically eliminated, followed by re-fitting the model and reassessing performance. This iterative procedure continues until the pool is reduced to a set between zero and the total number of environmental covariates. This method relies on out-of-bag (OOB) cross-validation, making it a reliable selection approach for models such as

random forests, even though it does not test every possible combination of covariates (Nussbaum et al., 2018). The RFE process is independently conducted on each subset, leveraging the default hyperparameters of the random forest algorithm as provided by the *RandomForestRegressor* package in Python. The optimal subset of variables is identified when further iterations no longer yield improvements in model performance, defined by the minimization of the loss function. For this study, the OOB root-mean-square error (RMSE) was used as the loss function. The ultimate set of covariates was identified as the combination that minimized the

loss function. The aforementioned analysis was executed for all target variables and depths. For instance, with surface (0-5 cm) soil organic carbon, 35 environmental covariates remained for analysis after the filtering process (Fig. 3), and marked with a superscript "1" in Table S1.

## 2.3 Digital soil mapping

### 2.3.1 Spatial prediction and uncertainty

The Random Forests (RF, (Breiman, 2001)) and Quantile Regression Forest (QRF) model were employed to evaluate the statistical relationship between each soil property at six layers and environmental covariates. The RF model in this study was used to generate mean predictions, while QRF were applied to produce prediction maps at different quantiles, providing a more comprehensive representation of uncertainty. The QRF algorithm, introduced by Meinshausen, (2006), is an ensemble machine learning model that utilizes tree structures and bootstrapping techniques to create a collection of tree models. Each tree is developed

from a learning set generated by repeatedly sampling calibration samples through bootstrapping, with node splits influenced by a randomly selected subset of covariates. The final prediction value at each predetermined quantile is obtained by averaging the predicted values from all trees. Building on the foundation of RF, QRF algorithm present a novel approach to enhancing regression

tree performance (Koenker, 2005). In RF, averaging across multiple tree-based models results in more accurate predictions compared to using a single regression tree. The QRF offers insights into the full conditional distribution of the dependent variable.

Consequently, conditional quantiles can be inferred using QRF algorithm. The conditional distribution of $Y$ given $X = x$ is defined as $F(y|X = x) = P(Y \leq y|X = x)$. To estimate $F(y|X = x)$, a weighted empirical cumulative distribution function is considered:

$$\hat{F}(y|X = x) = \sum_{i=1}^{n} w_i(x, \theta) Y_{\{Y \leq y\}} \tag{1}$$

The tree-based model developed using QRF algorithm follows the RF methodology. However, unlike RF, where only the mean of the observations within each node is retained, the QRF approach preserves the values of all observations within each node. This

comprehensive set of observations in each node is utilized to derive the quantiles, which are subsequently used to construct prediction intervals. These intervals serve as a measure of the prediction uncertainty, providing a more detailed understanding of the conditional distribution of the target variable. Additionally, the uncertainty estimates evaluated by QRF are likely more accurate and interpretable than those derived from regression kriging, particularly in areas with sparse samples (Liu et al., 2022a). Furthermore, RF and QRF are capable of handling complex non-linear relationships and multivariate interactions, offering high

predictive power (Gyamerah et al., 2020). This distinguishing advantage sets RF and QRF apart from other machine learning algorithms (Liu et al., 2022b).

Separate models were developed independently for each soil layer, ensuring no overlap of observations from the same profile across training and testing datasets. The selection of hyper-parameters, specifically the number of randomly selected variables from all predictors (*max_features*) and the minimum node size (*min_samples_leaf*), plays a crucial role in determining the performance

of the RF model. These hyper-parameters significantly influence the model's predictive accuracy. Other parameters, such as the number of trees (*n_estimators*), were not optimized during the RF's training process. To address potential overfitting concerns, the values of *max_features* and *min_samples_leaf* were fine-tuned using a 10-fold cross-validation method. This approach involved randomly dividing the training dataset into ten folds. One-tenth of these sub-datasets was utilized as the validation sample, while the remaining sub-datasets were applied for training the RF and QRF model. This tuning was conducted using the gridded direct

search approach, with *max_features* explored within the range of [1, 30] at single intervals, and *min_samples_leaf* within the range of [5, 30] at intervals of five. In this study, the aforementioned hyperparameter search was conducted for each of the six soil depth layers for every soil property. These hyperparameters were then used for modeling and spatial prediction of the corresponding soil property variables at their respective depths. To maintain brevity, Table S2 presents the tuned model hyper-parameters for each soil property considered at the 0-5 cm depth interval.

The relative importance of covariates in the trained RF and QRF model were assessed to investigate the impact of environmental factors on spatial variations of soil properties. This importance was determined by evaluating the influence of each covariate on the model's prediction performance. The relative importance of each covariate was quantified using the increase in mean square error (%IncMSE), a metric derived from permuting the values of a covariate to remove its information content. By comparing the model's accuracy before and after permutation, it was possible to determine how crucial each covariate was in

predicting soil properties. A higher %IncMSE indicated a greater importance of the covariate, signifying that its presence

substantially contributed to the model's predictive accuracy. This relative importance allows for a detailed analysis of how different environmental factors control spatial variations in soil properties, providing valuable insights for digital soil mapping.

Mapping China, covers approximately 9.6 million km², at 90 m resolution requires more than $10^9$ pixels for each soil property at each depth, posing a considerable challenge. Due to the extensive geographic coverage and high-resolution requirements in soil mapping for this study, predicting each soil property at a specific depth involves a substantial volume of data, with environmental covariates data reaching up to 470 GB. Faced with such extensive data processing demands, conventional single-machine resources often prove inadequate and challenging to cope with. Therefore, to overcome the memory limitations imposed by high-resolution mapping and enhance the computational efficiency of spatial prediction, we implemented parallel computing. Initially, we partitioned environmental covariates into distinct 1°×1° tiles. Using the finalized model, a single core performed spatial predictions within each block. Leveraging multiple cores processing, we simultaneously handled multiple tiles, significantly accelerating spatial predictions. Upon acquiring the outcomes for every tile, we utilized image mosaicking to seamlessly integrate these outputs, ultimately assembling the comprehensive map of various soil properties and depths across China. All the experiments are performed on a Linux server with Intel Core (TM) i9-10980XE, 3.00GHz×64 CPU, 512 GB RAM (Random Access Memory) and two NVIDIA RTX A5000 graphics cards. All scripts were written in the open-source Python programming environment with Python version 3.11.4 (https://www.python.org/) using PyCharm with version 2024.3.28. The "RandomForestQuantileRegressor" and "RandomForestRegressor" package were employed for model construction. The optimization of the model was performed using "scikit-learn" library, while the "gdal" and "matplotlib" packages were utilized for data processing and visualization, respectively.

Using the selected environmental covariates from the aforementioned feature engineering, the constructed models were applied to compute four statistical values—mean, 0.05 quantile ($q_{0.05}$), median (0.50 quantile, $q_{0.50}$), and 0.95 quantile ($q_{0.95}$)—at every 90 m pixel across all standard depth layers (0-5, 5-15, 15-30, 30-60, 60-100, and 100-200 cm) specified by GlobalSoilMap (Arrouays et al., 2014) over China, capturing the conditional distribution of soil properties. Although the performance differences between mean predictions using RF and median predictions using QRF are minimal, their ability to capture extreme values (i.e. both high and low values) was considered. In this study, we evaluated the performance of RF and QRF models by not only the overall statistical metrics but also their capacity to predict extreme values, to determine the most suitable model for generating national gridded soil maps of various soil properties at a 90 m resolution. As shown in Table S7, soil properties such as soil pH, silt, clay, TP, Red (R) of wet soil color, Blue (B) of wet soil color, Red (R) of dry soil color, and Blue (B) of dry soil color were modeled using median predictions from QRF, as this approach better captured extreme values. Similarly, the study by Helfenstein et al., (2024) also assessed mean predictions by RF and median predictions by QRF, highlighting that for certain soil properties, median predictions are more appropriate than mean predictions. For most other soil properties in this study—such as sand, BD, OC, gravel, AN, TN, CEC, porosity, TK, AK, AP, Green (G) of wet soil color, and Green (G) of dry soil color—mean predictions from RF were used to generate the 90 m resolution soil maps. The better model was consistent across different depths for the same soil property; thus, Table S7 only presents the performance comparison of mean and median predictions for the surface layer (0-5 cm depth interval) and either the mean or the median is used for the mapping of a soil property for all depths. The calculated median, along with the 0.05 and 0.95 quantiles, was also used to estimate uncertainty. Uncertainty was expressed as the upper and lower limits of a 90%

prediction interval, represented by the empirical distribution's 0.05 and 0.95 quantiles, respectively. Furthermore, to facilitate comparison, the prediction interval relative to the median ($q_{0.50}$) was used as an indicator of uncertainty (Liang et al., 2019; Liu et al., 2022a). A higher ratio for a pixel indicates greater uncertainty in the predicted value for that location (Poggio et al., 2021). When developing the 90 m resolution soil maps in this study, either mean or median predictions were selected for storage efficiency. However, for lower-resolution maps provided at 1 km and 10 km, in addition to mean and median predictions, we also included

prediction maps for the 0.05 and 0.95 quantiles. These additional maps are helpful for illustrating data uncertainty.

For the sand, silt, and clay content from the FNSSC and SNSSC, they were measured following the schemes of International Society of Soil Science (ISSS) and Katschinski (Katschinski et al., 1956). Since most land surface models (LSMs) and other applications require soil texture data in the FAO-USDA system, we used several particle-size distribution models (Shangguan et al., 2013) to convert the original ISSS and Katschinski particle-size distribution data into the FAO-USDA system. A 5% quality control

threshold was applied, excluding soil profile samples where the sum of the three fractions fell outside the 95%‑105% range (Shangguan et al., 2013), and they were converted to make sure the sum of them are 100% by using the weighting approach. For the mapping of each particle size fraction (sand, silt, and clay), separate spatial prediction models were developed, and the weighting approach was applied to ensure that the sum of the three fractions equaled 100%.

### 2.3.2 Evaluation criteria

To validate the performance of RF and QRF model for generating CSDLv2, two validation methods were employed to ensure that the CSDLv2 product has low errors in both spatial and vertical depth scales against laboratory measurements values. The first method involved randomly selecting 10% of the multi-source soil profiles as test samples, while the remaining 90% were used for training the model (i.e., data-splitting). The second method took the WoSIS dataset as an external independent validation dataset, with the rest of the data used for model training (i.e., independent samples). We choose WoSIS as the independent validation dataset

because it has a spatial distribution close to a probability sampling (Brus et al., 2011). Based on the training soil profiles, these two validation approaches were implemented to assess the accuracy performance of predictive mapping for each soil property at various depths. Three statistics, namely, mean prediction error (ME), root mean square prediction error (RMSE), and Modelling Efficiency Coefficient (MEC, (Krause et al., 2005)) were calculated to evaluate the models' predictive performance. They were calculated as follows:

$$ME = \frac{1}{N}\sum_{i=1}^{N} \varepsilon(s_i) \tag{2}$$

$$RMSE = \sqrt{\frac{1}{N}\sum_{i=1}^{N} \varepsilon(s_i)^2}, \tag{3}$$

$$MEC = 1 - \frac{\sum_{i=1}^{N}(z(s_i)-\hat{z}(s_i))^2}{\sum_{i=1}^{N}(z(s_i)-\bar{z})^2}, \tag{4}$$

, where $z$ represents the observed soil variable, $\hat{z}$ is the predicted soil variable at location $s_i$ ($i = 1, …, N$; $s_i \in \wp$ ), and $N$ is the total number of population units in the study area $\wp$. Regard the prediction error as the difference between the observed ($z$) and predicted

($\hat{z}$) values of a soil property at the $i^{th}$ spatial location, denoted by $\varepsilon(s_i) = z(s_i) - \hat{z}(s_i)$. To guarantee the accuracy and reliability of our results, we performed 20 repetitions of 10-fold cross-validation and calculated the mean and standard deviation of the measurements.

     The soil property maps predicted in this study were compared to three existing soil map datasets. The first dataset is SoilGrids 2.0, accessible at https://soilgrids.org/, which has a 250 m resolution (Poggio et al., 2021). It represents an advancement over
previous global soil properties maps, known as SoilGrids250m (Hengl et al., 2017), incorporating the up-to-date machine learning methods and benefiting from the expanded availability of standardized soil profile data worldwide, along with environmental covariates (Poggio et al., 2021). The second dataset is CSDLv1 with a resolution of 1 km (Shangguan et al., 2013), accessible at http://globalchange.bnu.edu.cn. Lastly, we considered the Harmonized World Soil Database v2.0 (HWSD 2.0), known for its soil property maps created via a soil type linkage method, available at https://www.fao.org/soils-portal/data-hub/soil-maps-and-
databases/harmonized-world-soil-database-v20/en/.The HWSD 2.0 has been synthesized by integrating regional and national soil data globally (FAO & IIASA, 2023). To quantify the enhancement of our predictions over existing soil maps, we calculated the relative improvement ($RI$) using both RMSE and MEC metrics, employing the following equations:

$$RI_{RMSE} = \frac{RMSE_{existing} - RMSE_{CSDLv2}}{RMSE_{existing}} \tag{5}$$

$$RI_{MEC} = \frac{MEC_{CSDLv2} - MEC_{existing}}{MEC_{existing}} \tag{6}$$

, where $RI_{RMSE}$ and $RI_{MEC}$ denote the relative improvement concerning $RMSE$ and $MEC$, respectively. $RMSE_{new}$ and $MEC_{new}$ represent the accuracy statistics for predictions in this study, while $RMSE_{existing}$ and $MEC_{existing}$ signify the accuracy statistics for the existing soil maps. An $RI > 0$ denotes CSDLv2 outperforms the existing soil maps.

     Considering the unavoidable impact of various error sources on any model for DSM, it is essential to quantify the associated mapping uncertainty (Lilburne et al., 2024; McBratney et al., 2018). To evaluate uncertainty, the prediction interval coverage
probability (PICP) was employed based on the randomly held-back soil profile test samples. PICP represents the proportion of observations at each depth encapsulated by the corresponding prediction interval (Li et al., 2023). In this study, the prediction interval was estimated using the aforementioned QRF model. If the uncertainty estimates are reasonably defined, the PICP should yield an estimate of 90% for a 90% (or 0.9) prediction interval. A PICP significantly greater than 0.9 suggests that the uncertainty has been underestimated, whereas a PICP significantly less than 0.9 indicates that it has been overestimated (Liu et al., 2020; Poggio
et al., 2021).

## 3 Results

### 3.1 Statistical analysis

     The probability density distributions of topsoil (0-5 cm) properties from different data sources are shown in Fig. S1, with different colors representing different data sources. If a color representing a data source is absent in some probability density

distribution charts, it indicates that the soil property is not available from that data source. As observed in Fig. S1, the probability density distributions of soil properties from multiple sources exhibit a generally similar trend, with minor differences that increase the spatial representativeness of the soil profile samples, rather than representing specific soil types. The abundance of soil profile data allows for a more detailed characterization of spatial variations in soil properties, particularly in a large and topographically diverse country like China (Liu et al., 2022a). Descriptive statistical analyses of soil properties across six standard depths are presented in Table S3. For most soil property variables at multiple depths, there is an extensive amount of soil profile data. Different soil properties exhibit varying trends with depth, accompanied by a large range and variation (see coefficient of variation). The vertical changes in soil properties vary depending on the specific soil property and soil type. For example, the content of OC and TN generally decrease with increasing depth in most soil types, exhibiting positive skewness distributions. However, other properties, such as soil pH or BD, show different vertical patterns depending on soil composition and local conditions. Regarding the homogeneity of variance, Levene's test between samples from different depths yielded p-values greater than 0.05 for soil property, indicating no statistically significant differences between samples from different depths.

## 3.2 Predictive performance

After training and optimization, the effectiveness of the RF and QRF model was evaluated. Using the test set, the model's prediction accuracy across multiple depths was assessed under two validation methods: Table 3 and Table S4 presents the predictive performance using a data-splitting strategy, where 10% of aggregated soil profiles were randomly partitioned as the test set. This validation of CSDLv2 was compared with the validation of the three existing soil map datasets using all soil profiles in this study. Table S5 displays the model's performance when modeling soil profiles from remaining data sources, validated independently using WoSIS data.

Overall, model performance varied depending on soil properties. The mean ME values were nearly zero, indicating that the predictions were generally unbiased. Soil pH was predicted with the highest accuracy, with MEC performance ranging from 0.75 to 0.68 across depths in the data-splitting validation strategy. That is to say that more than 68% of the pH variation can be explained and the predicted values are in good agreement with the laboratory measurements values. This result is consistent with previous studies (Chen et al., 2019; Hu et al., 2024; Lu et al., 2023). The mean MEC for sand and clay content were slightly higher than those for silt content, indicating that sand and clay are slightly more predictable than silt. As soil depth increased, MEC values showed a decreasing trend, while RMSE values increased, suggesting a vertical decline in the predictability of soil texture. This decline may be attributed to the fact that environmental covariates primarily reflect surface conditions, leading to reduced correlation with deeper soil properties. Additionally, the decrease in sample size at greater depths may also contribute to this trend. Similar observations have been noted in other related studies (Liu et al., 2020; Poggio et al., 2021). The model's predictive performance at the 5-15 cm depth interval was better than at the 0-5 cm depth interval, with higher MEC values and lower RMSE values. The prediction accuracy for OC was relatively high, with approximately 25% to 60% of the variation in OC across all depth layers explained in both data-splitting and independent validation methods. This performance surpasses the accuracy reported in related literature for OC prediction (Liang et al., 2019; Padarian et al., 2017). The prediction accuracy for soil properties content such as BD, gravel, TN,

CEC, TK, and TP is higher at depths less than 30 cm. These models can explain 30% to 60% of the variation in these soil properties, with accuracy comparable to that reported in related studies (Mulder et al., 2016; Ramcharan et al., 2018).

The model's performance varied with soil depth. For most soil property variables, including OC, TN, and BD, predictive accuracy decreased significantly with increasing depth. In contrast, the accuracy for CEC, gravel content, and TK only slightly declined. This decrease in accuracy for deeper layers has been noted in previous studies on soil organic carbon prediction (Mulder et al., 2016; Padarian et al., 2017), primarily because most environmental covariates predominantly characterize surface conditions, leading to weaker correlations with deeper soil layers (Liu et al., 2020). Conversely, the prediction accuracy for soil pH slightly

increased with depth. This improvement may be partly due to the increased stability of soil pH in deeper layers over large areas, leading to more consistent relationships with environmental factors (Liu et al., 2020). This observation aligns with the findings of Padarian et al., (2017). Additionally, independent samples validation is an effective approach to assess the validity of models and has been utilized in multiple studies (Lamichhane et al., 2019). Table S5 summarizes the model's predictive performance based on independent validation and compares it with other data products. These results also demonstrate the reliability of the predictive

model.

## 3.3 Spatial patterns

Fig. 4 illustrates the maps of soil physical and chemical properties, at the soil surface (0-5 cm) over China at 90 m resolution. The spatial distribution of the complete soil properties (as listed in Table 1) can be found in the Fig S2-24.

As shown in Fig. 4(a), the pH values ($H_2O$) in the topsoil range from 4.3 to 9.8. Soils south of 30°N are predominantly acidic

to strongly acidic, while those in the northern and northwestern regions are mostly basic or strongly basic. In some southern hilly and northeastern forested areas, soils appear to be acidic (pH < 7). In certain northern regions, especially in desert areas, soils are alkaline (pH > 7). This distribution aligns with the common understanding that areas with low precipitation tend to have alkaline soils, whereas areas with high precipitation tend to have acidic soils.

As shown in Fig. 4(b) for BD, northern regions tend to have higher bulk density due to low organic matter content and frequent

agricultural activities. Southern regions generally have lower bulk density owing to higher organic matter content and higher porosity. Northwest arid regions exhibit high bulk density, while the Qinghai-Tibet Plateau has low bulk density. Southeastern coastal areas show significant variation in surface bulk density, heavily influenced by land use practices.

As shown in Fig. 4(c), the spatial predictions of OC content reveal significant regional differences. The highest OC levels are found in the eastern Tibetan Plateau, northeastern China, and northern Xinjiang, where human activities are minimal. In contrast,

the lowest OC content is observed in the northwestern desert regions. OC content shows a decreasing trend from southeast to northwest, corresponding to the influence of the southeast monsoon. OC content is closely related to climatic conditions and land use practices (Zhang et al., 2023b; Zhou et al., 2019b). The spatial pattern of total nitrogen (TN) is similar to that of OC content. Areas with high precipitation and good vegetation cover tend to have higher OC and TN levels, while areas with low precipitation and poor vegetation cover tend to have lower OC and TN levels. This is because both OC and TN are closely related to organic

matter input from vegetation. In regions with high vegetation productivity, organic matter contributes to both carbon and nitrogen accumulation in the soil, resulting in similar spatial patterns for OC and TN.

The mean predicted maps of soil texture (clay, silt, and sand contents) at different depths across China are shown in Fig 5(e)-(h), respectively. Overall, clay content was predicted to be low in the northern and northwestern regions, while higher in the southern regions. The lowest clay content was found in the deserts of the northwest, and the highest in the Yunnan-Guizhou Plateau. 425 Relatively higher clay content was observed in some southern provinces such as Guangdong and Guangxi. Silt content was predicted to be high in the Loess Plateau and eastern China, while it was lower in the deserts of the northern and northwestern regions. These findings were consistent with previous studies (Liu et al., 2020). The predicted soil texture patterns fit well with the general characteristics and distribution of known Chinese soils (Gong et al., 2014).

For CEC, the spatial distribution of surface CEC is shown in Fig 4(j). CEC represents the total amount of exchangeable cations 430 that soil can absorb, serving as a crucial indicator of soil fertility, nutrient retention capacity, and buffering capacity, thereby influencing plant growth. Lower CEC value indicate that the soil can store fewer nutrients. The CEC levels are closely related to soil type, climatic conditions, and land use practices (Beillouin et al., 2022). Generally, soils with higher clay and organic matter content have higher CEC values. Fig 4(j) indicates that higher surface soil CEC values are found in the Qinghai-Tibet Plateau and the peat and forest regions in the northeast (i.e., high-biomass or low-leaching areas). Lower CEC values are observed in the 435 southeastern regions and the arid and semi-arid areas in the north, with the lowest CEC values found in desert areas. The relatively low CEC in the southeastern regions is attributed to higher temperatures and rainfall, leading to strong leaching loss of exchangeable substances.

The spatial distribution of TK, TP, and AK are shown in Fig. 4(l), Fig. 4(m), Fig. 4(n), respectively. Sedimentary rocks in Southwest China are abundant in phosphorus, leading to relatively higher TP levels in soils derived from these rocks. In contrast, 440 South China's soils typically exhibit lower TP levels due to extensive weathering and leaching. Alpine regions with significant organic matter accumulation are predicted to have relatively high TP content. The concentrations of both TK and AK diminish generally from north to south, despite their distribution patterns are rather different. Low levels of TK are found in tropical regions, whereas high levels are located in the Qinghai-Tibet Plateau and northeastern China. High values of AK are dispersed throughout western Tibetan Plateau. The spatial patterns of the variables of interest listed in Table 1 at multiple depths can be found in the 445 supplementary materials (Fig. S2-S26). These spatial distributions are consistent with those reported in other similar studies (Hu et al., 2024; Liu et al., 2022a, Poggio et al., 2021).

**3.4 Prediction uncertainty**

Table S6 lists PICP values for different soil properties at multiple depths, calculated based on randomly held-back test samples. For a 90% (or 0.9) confidence interval, 90% of the observations are expected to fall within the predicted lower and upper limits. It 450 can be seen that the PICP values for all soil properties at six standard depths are very close to 90%, indicating that the predicted lower and upper limits estimated by the ensemble machine learning method are appropriate. In other words, the uncertainty estimates are largely reliable. It was observed that different soil properties exhibit distinct spatial patterns of prediction uncertainty, but

different depths of the same soil property show similar patterns. The accuracy assessment in Fig. S2-S24 shows the uncertainty maps of soil property predictions. For OC, regions with relatively simple terrain, such as deserts, the North China Plain, and the Northeast Plain, exhibit lower uncertainty. In contrast, the central Qinghai-Tibet Plateau and western Inner Mongolia, where sampling is sparse and OC content is low, show higher uncertainty. The Altai region, with its complex terrain and diverse landscape types, also exhibits relatively high uncertainty. For soil pH, regions with high prediction uncertainty are found in Southwest China, where samples are sparse in complex soil landscapes. As soil depth increases, the uncertainty in predictions for properties like OC and pH generally decreases due to the more stable nature of subsurface layers, reduced influence from external factors, and the fact that deeper soils are less affected by environmental covariates. Additionally, while topsoil is more complex and variable due to its interaction with the environment, subsurface layers tend to have more consistent properties, leading to a less uncertainty in predictions at depth (Liu et al., 2022a).

## 3.5 Relative importance of predictors

The relative importance of environmental covariates for soil properties prediction at the 0-5 cm depth interval is shown in Fig. 6 and Fig. S26, displaying only the top 15 most important environmental covariates. Overall, organism-type covariates account for a significant proportion among different categories of environmental factors. There are variations in the relative importance of environmental covariates across different soil property variables.

For soil pH, in the optimal ensemble machine learning model, the climate factor (eg. MODCF) was identified as the most important variable, with an importance exceeding 30%, significantly higher than other covariates. The leaf area index (LAI) ranks second in relative importance. Previous studies have also indicated that LAI is a key factor in predicting soil pH (Sun et al., 2023). Other environmental covariates had relatively smaller contributions. In terms of covariates types, organisms factors accounted for 50% of the contribution to soil pH prediction, followed by relief factors (23.9%) and climate factors (17.4%).

For OC content, terrestrial ecosystems (eg., TERECO) and climate factors (eg., MODCF) are the most important covariates, followed by depth to bedrock and elevation (DEM). Shallow bedrock typically results in thinner soil layers, which can limit soil development and the accumulation of OC. However, soils developed on shallow bedrock do not always have low OC, as the rate of OC accumulation can be significantly influenced by the type of vegetation present. In contrast, deeper bedrock allows for thicker soil layers, providing more space and time for OC accumulation. DEM can indirectly reflect differences in land use and vegetation types, which can also affect the distribution of OC content. This indicates that the prediction of soil organic carbon is influenced by multiple factors. Many studies have shown that organisms factors (e.g., landuse) is the most important predictor (Gomes et al., 2019).

For sand prediction, elevation and Mean Annual Cloud Frequency (eg., MODCF) rank as the top two most important covariates in the ensemble machine learning model. Altitude primarily affects soil through gravitational and erosional processes, which transport fine particles and leave behind coarse particles (Li et al., 2023). This is evident in the relatively higher sand content in most mountainous areas compared to adjacent lowland regions. Thermal processes drive physical weathering, while wind, water, and terrain govern erosion processes, predominantly shaping the distribution patterns of sand in China.

For silt prediction, climate-related factors (e.g., TNSMOD, MODCF, and wc2.1_srad) are the most important covariates. Apart from climate, terrain factors (e.g., DEM, DEM_vbf, and slope) also play crucial roles in silt prediction. Terrain features largely determine gravitational and hydraulic conditions, thereby influencing the erosion, redistribution, and sorting processes of soil particles. This observation is consistent with previous studies (Hengl et al., 2017), indicating that climate data can enhance the predictive performance of soil texture models.

For clay prediction, organism-type covariates (e.g., TERECO, Table S1) ranks as the most important environmental covariate, followed by climatic variable wc2.1_srad. Terrain-related variables (e.g., DEM, DEM_popn, and slope) rank second in importance overall, exerting their influence by controlling local moisture and thermal conditions, as well as redistributing terrain material (Liu et al., 2020). Other studies have similarly shown that vegetation indices, rock type, bioclimatic zones, and agricultural indices can help characterize changes in soil clay content (Ge et al., 2019; Hengl et al., 2017).

For CEC prediction, the most important covariate is terrestrial ecosystems (i.e. TERECO). Plant roots can alter the chemical environment of the soil by secreting organic acids and other substances, which influence the dissolution and reprecipitation processes of soil minerals. These changes can affect the soil's CEC. Shiri et al. (2017) investigated the relationships between soil carbon content, clay content, and particle size with CEC. They found that higher organic carbon and clay content significantly enhance CEC due to their high specific surface areas and cation retention capacities. This is consistent with our findings, where areas with higher organic content, influenced by plant root activity, showed higher CEC value. The relative importance of the top 15 environmental covariates for other soil properties across all depths is visualized in Figures S31-S52.

## 4 Discussion

### 4.1 Comparison with previous products

Table 3, S4 and S5 present the accuracy assessments of our predictions (i.e., CSDLv2), CSDLv1 (Shangguan et al., 2013), SoilGrids 2.0 (Poggio et al., 2021), and HWSD 2.0 (FAO & IIASA, 2023) at six standard depth intervals using data-splitting validation and independent sample validation methods, respectively. Table 3 lists the validation accuracy of selected soil properties with the highest prediction accuracy using the data-splitting validation method, while Table S4 provides the complete accuracy assessments for all soil properties of interest. Table S5 identifies the variables for which the WoSIS database can serve as independent samples. Overall, our predictions, whether using data-splitting validation or independent sample validation, achieved relatively higher MEC values and lower RMSE values across multiple depths for most target variables, demonstrating much greater accuracy than existing soil property maps (FAO & IIASA, 2023; Poggio et al., 2021; Shangguan et al., 2013; Song et al., 2020; Zhou et al., 2019b). Specifically, using data-splitting validation as an example, our predictions for pH demonstrated an absolute improvement in the mean MEC for all layers, increasing from 0.60 to 0.70, while the RMSE decreased from 0.77 to 0.68 compared to SoilGrids 2.0. In comparison to CSDLv1, our prediction performance for pH improved from 0.44 to 0.70, with the RMSE reducing from 0.96 to 0.68. Compared to HWSD 2.0, the prediction performance showed the greatest improvement in MEC and the most significant reduction in RMSE. The ME values indicated that SoilGrids 2.0 significantly overestimated TN content, whereas

CSDLv1 and HWSD 2.0 underestimated it. Additionally, in the independent validation (Table S5), across predictions of various soil properties at different depths, this study demonstrates overall predictive performance that is comparable to or better than SoilGrids 2.0, even though SoilGrids 2.0 used all the soil profiles of WoSIS in its production. Moreover, it shows superior performance compared to CSDLv1 and HWSD 2.0.

Such a national-scale publication of soil maps hides most of the details. Nevertheless, because the soil properties are predicted at a 90 m resolution, portions of the maps can be enlarged to reveal increasingly detailed information up to the limit of that resolution. Using the example of surface (0-5 cm) OC content, Fig. 5 shows a visual comparison within a window in western Sichuan Province (102.92°-104.08°E and 30.92°-32.08°N). This window corresponds to the red window in Fig. 2(a). The comparison is between the dataset developed in this study (CSDLv2) and the widely used SoilGrids 2.0, CSDLv1, and HWSD 2.0. The OC map produced in this study clearly reveals spatial variability with local morphology and provides more detailed information than the other three maps. Moreover, the CSDLv2 and SoilGrids 2.0 datasets, both products of advanced digital soil mapping techniques, exhibit notably higher OC content compared to the other two datasets generated through the linkage method across the majority of this region. This finding aligns well with our understanding of the area's environmental conditions: the cold climate at high elevations (Fig.5, DEM), coupled with extensive forest and grassland covers (Fig.5, Landuse), creates an ideal setting for the accumulation of OC in the soil. Fig. S26-30 shows the spatial details of other soil properties, including TN, gravel, porosity AN, and AP. Therefore, the fine soil property map with a spatial resolution of 90 m can better present the spatial variability of soil properties in related research, which can aid precision agriculture and soil management.

To characterize the spatial pattern differences between CSDLv2 and CSDLv1, Fig. 7 (a, c, e) illustrates the spatial difference maps of OC, sand, and clay predictions in CSDLv2 subtracted by those in CSDLv1 as an example. For OC, the differences are mainly observed in the Tibetan Plateau, Yunnan-Guizhou Plateau, and Northeast Plain, where OC content is higher in CSDLv2 than in CSDLv1. For sand, CSDLv2 shows relatively lower sand content in desert and semi-desert areas (e.g., Taklamakan Desert), while relatively higher sand content is observed in southern coastal regions. For clay, an opposite trend to sand is observed. The possible cause of these differences may be attributed to the linkage method used in developing CSDLv1, which averaged all soil profiles for a given soil type or soil polygon, neglecting local spatial variation in soil properties. Additionally, as shown in Fig. 5, the two datasets derived by DSM technology (i.e., CSDLv2 and SoilGrids 2.0) had similar spatial pattern and higher values than the other two, indicating an underestimation of OC content by the linkage method in this region. The scatter plots in Fig 7 (b, d, f) show the comparison between CSDLv2, CSDLv1, and the observed data. From the bivariate kernel density estimates and correlation coefficients, it is evident that CSDLv2 has a stronger correlation with the observed data. It can also be seen that the scatter points for CSDLv1, based on the linkage method, are more dispersed, whereas the scatter points for CSDLv2, based on DSM technology, are more concentrated. Compared to CSDLv2, CSDLv1 had a significant underestimation of OC and both significant overestimation and underestimation of sand and clay. This may be due to the better fitting ability of DSM technology with the available data. However, the use of the ensemble learning algorithm, which averages predictions from multiple trees, tends to smooth out extreme values during spatial extrapolation, potentially reducing variability in certain regions. On the whole, CSDLv2 provides a more accurate estimation of soil properties than CSDLv1, thus it may have significant influences on land surface modeling due to their

large differences in spatial distribution. Further studies are needed to demonstrate the impact of the new soil dataset compared to the old version and global soil datasets by running a land surface model (Li et al., 2020).

Based on the experimental results and analysis, compared to CSDLv1, the main advantages of CSDLv2 include the following aspects. First, CSDLv2's spatial resolution is 90 m, aligning with the resolution of the most important input layers used for the predictions, and this is an improvement over CSDLv1's 1 km resolution. This addresses the long-standing issue of lacking detailed and accurate soil information and enhances modeling of energy, water, and momentum processes in the land surface model. Second, high-resolution environmental covariates related to soil formation were used with advanced machine learning algorithms, replacing traditional soil transformation rules. In recent years, digital soil mapping technology has made significant progress, particularly with the success of machine learning in large-scale spatial predictions (Poggio et al., 2021). Numerous studies have shown that advanced machine learning models typically have better predictive performance than simpler models (Yan et al., 2020). Third, an RGB soil color system (i.e., red, green, and blue) has been added, resolving the inconvenience of only having the Munsell color system in the first edition dataset. This addition enhances the visual representation of soil colors and allows for better integration with digital platforms, remote sensing applications, and computer displays (Al-Naji et al., 2021). Finally, global validation was conducted using data-splitting and independent samples, and prediction uncertainty was quantitatively provided using QRF, rather than merely offering quality control information. Compared to other related data products: CSDLv2 encompasses more than 20 comprehensive soil physical and chemical properties, whereas most existing studies focus on mapping one or several fundamental soil properties, lacking comprehensive soil properties data set products (Liang et al., 2019; Chen et al., 2019; Zhou et al., 2019a; Liu et al., 2022a; Liu et al., 2020). For instance, AN serves as an indicator of soil fertility, reflecting the potential release of organic nitrogen and ammonium nitrogen in the soil. AK reflects the potassium available for plant uptake, which is crucial for plant growth and development. The extensive soil information has significant applications across various fields. Additionally, another advantage of CSDLv2 over both CSDLv1 and other related data products is that a large number of soil profile samples from different data sources were collected, enhancing the spatial representativeness of the soil profiles. Sample size is a critical factor affecting model performance (Padarian et al., 2020).

## 4.2 Potential applications of CSDLv2

The national-scale high-resolution soil property maps developed in this study have significant potential for applications in land surface modeling and Earth system modeling. These models simulate interactions between the land surface, atmosphere, and biosphere, making accurate representation of soil properties essential for improving model performance and predictions. For instance, soil pH is crucial for nutrient solubility, while CEC indicates fertility and nutrient retention capacity in land surface modeling. In biogeochemical process modelling with land surface modeling, OC, TN, and TP are key parameters and prognostic variables. These soil nutrients can be calculated by running models for thousands of years until an equilibrium state is reached, a process known as model "spin-up" (i.e., warm-up period) (Dai et al., 2019b; Shangguan et al., 2013). However, the non-linear feedbacks in biogeochemical cycles make such "spin-up" time-consuming and less reliable for initializing soil nutrients. Therefore, this dataset can also serve as an important benchmark for initial or calibration variables.

Currently, many soil properties are not yet utilized in land surface model simulations, with only soil texture, OC, gravel and
BD being primarily used. However, more soil properties can theoretically be employed as initial variables in Earth system modeling. Each soil property plays an important role in both Earth system modeling and land surface modeling, and although some properties are not yet used, they hold significant potential for future applications. For example, soil albedo is significantly correlated with the Munsell soil color value (hue, value, chroma). In some Earth system models, parameters derived from pedotransfer functions are used directly as inputs rather than being calculated within the models.

Moreover, CSDLv2 offers extensive possibilities for research and applications across various fields, including climate change research and carbon cycling (Chen et al., 2023), as well as supporting the spatial delineation of management zones in precision agriculture (Piikki et al., 2017). Regarding soil pH, for agricultural departments and farmers, fine mapping of soil pH holds significant value in local and field land use planning and management, as different crops exhibit optimal growth in soils with varying pH ranges (Hu et al., 2024). For instance, rice thrives best in soils with pH levels between 6.0 and 7.5, whereas peanuts prefer soils
with pH levels between 5.6 and 6.0. Thus, precise soil pH maps provide essential information for agricultural zoning and management. Furthermore, due to the widespread applicability of soil information, CSDLv2 also holds potential applications in numerous other fields.

## 4.3 Limitations and Outlook

Some advances have been made in this study, but several limitations still need to be addressed in future efforts. First, remote
sensing imagery has been used globally for soil property mapping (Guo et al., 2022; Xia and Zhang, 2022). With the advancement of remote sensing technology, more and more high-spatial-resolution free data have become available. For example, Xia and Zhang (2022) found that using high-spatial-resolution GF-2 imagery improved soil property prediction accuracy compared to medium-resolution imagery (e.g., Landsat 8 and Sentinel-2 imagery). Therefore, future digital soil mapping work can focus more on integrating high-resolution remote sensing products, which can enable models to capture the complex statistical relationships
between soil properties and the environmental covariates at fine scales (Mulder et al., 2016).

Secondly, soil is a three-dimensional volume with property variability in all three dimensions. In this study, the vertical dimension of soil variability was modeled using spline interpolation. It is noteworthy that smoothing spline interpolation standardizes soil layer data, which is not error-free, but due to the lack of a "true" depth function for each soil profile (vertically dense samples), the standardization error cannot be quantitatively estimated (Liu et al., 2022a). Recent publications have considered
observation depth as a covariate (Hengl et al., 2017; Nauman and Duniway, 2019), creating a "3D" model, but some studies indicate that this approach may be overly simplistic or lead to consistency issues in the predicted depth sequences (Ma et al., 2021). This might be true for local datasets, where short-range spatial variability and vertical variability have similar magnitudes (Poggio et al., 2021). Further research is needed to assess the impact of using depth as a covariate on national datasets and models. Additionally, alternatives such as "3D" models or geostatistical models utilizing 3D spatial autocorrelation are worth exploring (Helfenstein,
2024).

Thirdly, in this study, approximately 150 covariates related to soil properties, topography, climate, biomes, lithology, land use, and existing soil maps were collected. By removing inter-variable correlations and using recursive feature elimination, approximately 40 optimal variables were selected to map soil properties across the country. However, the original environmental variables with a resolution of 90 meters did not play a significant role in variable selection or importance ranking. Several reasons may explain this. First, many studies have confirmed that soil properties (e.g., soil pH) are highly correlated with lithology (e.g., soil group and parent material) and climatic factors, especially at large scales (Hu et al., 2024; Lu et al., 2023). Topography downscaling methods can be used to prepare high-resolution climate covariates (Chen et al., 2024). However, fine and reliable maps of these factors are typically unavailable, especially at large spatial scales. Therefore, when introducing these factors to map soil properties, coarse-resolution raster data (e.g., 1 km) often have to be used (Liu et al., 2022a; Lu et al., 2023). Secondly, in this study, some covariates (e.g., elevation, and slope) with an original resolution of 90 meters are highly correlated with soil properties (e.g., soil pH). However, these factors are also highly correlated with other factors such as mean annual temperature and mean annual precipitation (Guo et al., 2022). These factors were removed by the recursive feature elimination algorithm when selecting the optimal variables because they were highly correlated with the already retained existing variables. This also led to the relatively lower importance of these factors in contributing to the models for soil properties (e.g., soil pH). Therefore, the final maps of soil properties with a 90-meter resolution in this study will be useful for practical decision-making. In future work, introducing fine-resolution environmental covariates is expected to improve mapping accuracy.

Last but not least, although this study utilized multi-source soil profile data from different time periods to develop comprehensive static maps of soil properties, the CSDLv2 maps mainly represent the status of soil in the 1980s, as most soil profiles come from the SNSSC. For soil properties that change over time, other multi-source soil profile data have not been fully utilized. Together with maps based on data from other period, such as 2010s from Liu et al., (2022a), CSDLv2 could provide new perspectives for studying temporal changes in soil properties. However, more efforts are needed to model the temporal change of soil properties with more time slices, especially for those soil properties which may change in short term. In this aspect, the undergoing Third National Soil Survey of China and other legacy soil profiles should be exploited to map time series of soil properties using spatial-temporal modelling technology. As CSDLv2 is developed on the national scale, the maps are suitable for broad-scale applications, such as national scale and a large regional scale (e.g., provincial-level) analyses. Although generated at a high resolution (90 m), they may not provide sufficient accuracy for farm- or field-scale applications, where locally calibrated models and detailed surveys are recommended. Users should consider the provided accuracy metrics and uncertainty maps to assess suitability for specific applications (Helfenstein et al., 2024).

## 5 Data and code availability

All resources of ensemble machine learning model, including training and testing code is publicly available at https://github.com/shgsong/CSDLv2. The soil maps in this study for six depth layers (0-5, 5-15, 15-30, 30-60, 60-100, and 100-200 cm) at 90 m spatial resolution across China are openly accessible https://www.scidb.cn/s/ZZJzAz or https://doi.org/10.11888/Terre.tpdc.301235 (Shi et al., 2024). Users can efficiently download the data sets provided in the first link

of the above statement by using the File Transfer Protocol (FTP) account information provided at the above links and common FTP client tools such as Filezilla (https://filezilla-project.org/) or FlashFXP (https://www.flashfxp.com/).

To meet the spatial resolution requirements of different applications, CSDLv2 not only provides soil properties at a 90 m resolution but also offers at 1 km and 10 km resolutions with maps of mean, median, 0.05 and 0.95 quantile. These 1 km and 10 km resolution data were derived from spatial predictions made by the constructed model using environmental covariates at the corresponding resolutions. The dataset is provided in raster format, available in both Network Common Data Form 4 (NetCDF4) and GeoTIFF (GTiff) formats.

## 6 Conclusions

The second version of the high-resolution national soil information grid for China was developed in this study, utilizing a vast number of multi-source legacy soil profile samples and advanced machine learning techniques, as a replacement for the first version dataset. This version includes over 20 soil physical and chemical properties, with prediction maps for each soil property covering six standard depths (0-5, 5-15, 15-30, 30-60, 60-100, and 100-200 cm). By combining ensemble machine learning with currently available high-resolution environmental covariates, the spatial variations of soil properties across China and at different depths can be effectively predicted. Overall, all the soil property maps performed well, accurately representing the spatial variations of soil properties. Under both data-splitting and independent samples schemes, CSDLv2 generally outperformed other gridded soil datasets, including CSDLv1, SoilGrids 2.0, and HWSD 2.0. CSDLv2 provided more spatial details and better represented the spatial variation characteristics of soil properties in China compared to other soil products. Furthermore, as this dataset is primarily based on legacy soil profiles from the Second National Soil Survey of China and describes the state of soil properties in the 1980s, it serves as a valuable complement to maps based on 2010s soil profiles, providing new perspectives for studying temporal changes in soil properties. These prediction maps also contribute to China's input to the GlobalSoilMap project and can be used for various hydrological, ecological analyses, and regional earth system modeling, especially for applications requiring high-resolution soil property maps. Future work can improve soil property mapping by employing advanced deep learning methods and incorporating more observations, particularly in regions with sparse samples like western China. Additionally, integrating high-resolution remote sensing data, developing more accurate 3D models, and accounting for temporal changes in soil properties will further enhance the mapping accuracy and usefulness of CSDLv2.

## 7 Author contributions

WSG conceived the research and secured funding for the research. GSS and WSG performed the analyses. GSS conducted the research and wrote the initial draft of the manuscript. WSG and GSS reviewed and edited the paper before submission. All other authors joined the discussion of the research.

## 8 Competing interests

The authors declare that they have no conflict of interest.

## 9 Acknowledgements

The authors are grateful to all the data contributors who made it possible to complete this research.

## 10 Financial support

This work was supported by the Natural Science Foundation of China (under Grants 42375144, 4227515 and 42205149), and Guangdong Major Project of Basic and Applied Basic Research (2021B0301030007).

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

**Table 1. List of Information of Soil Profiles Data**

| Soil property | Acronym | Units | Description | Maps |
|---|---|---|---|---|
| Bulk density | BD | g/cm$^3$ | Bulk density of the fine earth fraction oven dry | Figure S2 |
| Sand | sand | % | Gravimetric percentage of sand (2-0.05mm) in the fine earth fraction of the soil | Figure S3 |
| Silt | silt | % | Gravimetric percentage of silt (0.05-0.02mm) in the fine earth fraction of the soil | Figure S4 |
| Clay | clay | % | Gravimetric percentage of clay (< 0.02mm) in the fine earth fraction of the soil | Figure S5 |
| Rock fragment | gravel | g/100g | Volumetric content of fragments > 2 mm in the whole soil | Figure S6 |
| Porosity | porosity | cm$^3$/cm$^3$ | Volume fraction of void space (pores) in a material | Figure S7 |
| Wet color | R (Wet) | - | RGB quantified soil color for wet soil | Figure S8 |
|  | G (Wet) |  |  | Figure S9 |
|  | B (Wet) |  |  | Figure S10 |
| Dry color | R (Dry) | - | RGB quantified soil color for dry soil | Figure S11 |
|  | G (Dry) |  |  | Figure S12 |
|  | B (Dry) |  |  | Figure S13 |
| Wet color | Hue, value, chroma | - | Soil color of wet soil is represented by the Munsell notation with three dimensions: hue, value, and chroma | Figure S14 |
| Dry color | Hue, value, chroma | - | Soil color of dry soil is represented by the Munsell notation with three dimensions: hue, value, and chroma | Figure S15 |
| pH value (H$_2$O) | pH | - | Negative common logarithm of the activity of hydronium ions (H$^+$) in water | Figure S16 |
| Soil organic carbon | OC | g/100g | Gravimetric content of organic carbon in the fine earth fraction | Figure S17 |
| Cation exchange capacity | CEC | me/100g | Capacity of the fine earth fraction to hold exchangeable cations | Figure S18 |
| Total nitrogen | TN | g/100g | Total nitrogen in soil, comprising organic, inorganic, and ammonium nitrogen, among other forms | Figure S19 |
| Total phosphorus | TP | g/100g | Total phosphorus in soil includes all phosphorus compounds, both organic and inorganic, irrespective of their plant availability. | Figure S20 |
| Total potassium | TK | g/100g | Total potassium in a soil sample comprises both exchangeable (plant-available) and non-exchangeable forms. | Figure S21 |
| Alkali-hydrolysable nitrogen | AN | mg/kg | Total amount of nitrogen released from soil through alkali treatment (i.e. sodium hydroxide or potassium hydroxide) | Figure S22 |
| Available potassium | AK | mg/kg | Portion of potassium in the soil that is readily accessible for plant uptake | Figure S23 |
| Available phosphorous | AP | mg/kg | Fraction of phosphorus in the soil that is soluble in a chemical extract and readily accessible for plant uptake. | Figure S24 |

 **Table 2. Summary of the main high-resolution environmental covariates. For the complete list of soil forming factors, see Table S1.**

| Factors definitions | Description | Resolution (m) | Source |
|---|---|---|---|
| BDTICM | Depth to bedrock of China | 90 | http://globalchange.bnu.edu.cn/research/cdtb.jsp |
| B5/B7 | The ratio of Band 5 (near-infrared) to Band 7 (shortwave infrared 2) surface reflectance | 90 | https://www.usgs.gov/landsat-missions/landsat-collection-2 |
| NDVI | Normalized Difference Vegetation Index | 90 | Calculated from Landsat 8 Collection 2 Level-2 (LC08C02) on the GEE platform |
| NDWI | Normalized Difference Water Index | 90 | Calculated from LC08C02 on the GEE platform |
| surR | Surface Reflectance | 250 | https://modis.gsfc.nasa.gov/data/dataprod/mod09.php |
| EVI | Enhanced Vegetation Index | 90 | Calculated from LC08C02 on the GEE platform |
| SAI | Snow Area Index | 90 | Calculated from LC08C02 on the GEE platform |
| NPP | Net Primary Productivity | 500 | https://lpdaac.usgs.gov/products/mod17a3hgfv061/ |
| CanopyHeight | Canopy Height | 10 | https://doi.org/10.3929/ethz-b-000609802 |
| landcover | Land cover | 30 | http://www.sciencemag.org/content/342/6160/850 |
| Sentinel2 (B2, B3, B4, B8, B9) | Band2, 3, 4, 8, 9 from Sentinel2 | 30 | Derived from Sentinel2 on the GEE platform |
| QA_PIXEL | Landsat 8 Collection 2 Level-2 Pixel Quality Band | 90 | Derived from LC08C02 on the GEE platform |
| QA_RADSAT | Radiometric Saturation Quality control | 90 | Derived from LC08C02 on the GEE platform |
| SR (B4, B5, B6, B7) | Surface Reflectance of Band4, 5, 6, and Band7 | 90 | Derived from LC08C02 on the GEE platform |
| ST_ATRAN | Atmospheric Transmittance | 90 | Derived from LC08C02 on the GEE platform |
| ST_B10 | Band 10 Surface Temperature | 90 | Derived from LC08C02 on the GEE platform |
| ST_EMSD | Emissivity standard deviation | 90 | Derived from LC08C02 on the GEE platform |
| ST_TRAD | Thermal Radiance | 90 | Derived from LC08C02 on the GEE platform |
| ST_URAD | Downwelled Radiance | 90 | Derived from LC08C02 on the GEE platform |
| DEM | Land surface elevation | 90 | https://hydro.iis.u-tokyo.ac.jp/~yamadai/MERIT_DEM/ |
| slope | Terrain slope | 90 | Derived from DEM |
| Land use | Land use type | 30 | https://www.resdc.cn/DOI/DOL.aspx?DOIID=54 |
| RTMUSG15 | Rock type | 250 | https://doi.pangaea.de/10.1594/PANGAEA.788537 |

**Table 3. Accuracy evaluation of the selected soil properties with the highest prediction accuracy in CSDLv2, CSDLv1, SoilGrids 2.0, and HWSD 2.0, based on the randomly held-back soil profiles. The "Number" column indicates the number of samples used during testing. Refer to Table S4 for the complete accuracy evaluation of the soil properties considered. See Table 1 for the abbreviations and units of the soil properties of interest.**

| Property | Depth interval | Number | CSDLv2 | | | CSDLv1 | | | SoilGrids 2.0 | | | HWSD 2.0 | | |
|---|---|---|---|---|---|---|---|---|---|---|---|---|---|---|
| | | | MEC | RMSE | ME | MEC | RMSE | ME | MEC | RMSE | ME | MEC | RMSE | ME |
| pH | 0-5 | 830 | 0.69 | 0.70 | 0.00 | 0.48 | 0.92 | -0.03 | 0.60 | 0.79 | -0.15 | 0.35 | 1.03 | -0.28 |
| | 5-15 | 830 | 0.70 | 0.68 | 0.00 | 0.50 | 0.90 | -0.02 | 0.61 | 0.77 | -0.12 | 0.36 | 1.02 | -0.13 |
| | 15-30 | 822 | 0.70 | 0.68 | 0.00 | 0.26 | 1.21 | -0.41 | 0.60 | 0.77 | -0.16 | 0.38 | 1.03 | -0.15 |
| | 30-60 | 800 | 0.68 | 0.70 | -0.00 | 0.43 | 0.94 | -0.04 | 0.59 | 0.78 | -0.15 | 0.38 | 1.02 | -0.17 |
| | 60-100 | 648 | 0.68 | 0.70 | 0.00 | 0.44 | 0.94 | 0.04 | 0.59 | 0.78 | -0.14 | 0.39 | 1.01 | -0.18 |
| | 100-200 | 204 | 0.75 | 0.60 | 0.00 | 0.53 | 0.84 | -0.05 | 0.63 | 0.70 | -0.09 | 0.52 | 0.87 | -0.08 |
| sand | 0-5 | 874 | 0.67 | 12.15 | 0.05 | 0.19 | 22.19 | -2.24 | 0.60 | 13.08 | -1.84 | 0.20 | 21.84 | 2.38 |
| | 5-15 | 815 | 0.71 | 11.23 | 0.06 | 0.18 | 21.90 | -2.28 | 0.62 | 11.87 | -1.93 | 0.19 | 21.43 | 1.40 |
| | 15-30 | 812 | 0.71 | 11.41 | 0.05 | 0.15 | 22.58 | -1.67 | 0.62 | 11.85 | -1.71 | 0.14 | 21.89 | 2.63 |
| | 30-60 | 784 | 0.69 | 12.16 | 0.06 | 0.13 | 23.26 | -1.31 | 0.59 | 12.68 | -1.80 | 0.12 | 22.57 | 3.68 |
| | 60-100 | 638 | 0.68 | 12.85 | 0.04 | 0.11 | 23.22 | -1.30 | 0.51 | 13.53 | -1.94 | 0.10 | 23.45 | 4.03 |
| | 100-200 | 213 | 0.64 | 13.72 | 0.02 | 0.10 | 24.22 | -1.42 | 0.49 | 14.59 | -1.88 | 0.09 | 24.11 | 3.98 |
| silt | 0-5 | 893 | 0.61 | 9.81 | 0.02 | 0.11 | 16.78 | 2.02 | 0.55 | 10.54 | -0.58 | 0.10 | 17.38 | -4.44 |
| | 5-15 | 832 | 0.65 | 8.99 | -0.00 | 0.13 | 16.31 | 2.29 | 0.58 | 9.22 | -0.33 | 0.10 | 16.90 | -5.55 |
| | 15-30 | 830 | 0.67 | 8.76 | 0.00 | 0.13 | 16.29 | 2.12 | 0.60 | 9.02 | -0.51 | 0.09 | 17.30 | -6.46 |
| | 30-60 | 802 | 0.63 | 9.49 | 0.00 | 0.11 | 16.55 | 1.76 | 0.57 | 9.68 | -0.41 | 0.10 | 17.53 | -6.36 |
| | 60-100 | 656 | 0.62 | 10.08 | 0.00 | 0.10 | 17.05 | 1.49 | 0.55 | 10.34 | -0.33 | 0.10 | 18.07 | -6.15 |
| | 100-200 | 221 | 0.64 | 10.60 | 0.01 | 0.09 | 17.94 | 0.70 | 0.54 | 11.25 | -0.99 | 0.11 | 19.14 | -5.15 |
| clay | 0-5 | 914 | 0.63 | 6.74 | 0.01 | 0.12 | 11.23 | 0.21 | 0.52 | 7.60 | 2.49 | 0.12 | 11.14 | 2.06 |
| | 5-15 | 854 | 0.67 | 6.50 | 0.01 | 0.09 | 11.28 | 0.03 | 0.58 | 7.18 | 2.36 | 0.09 | 11.89 | 4.23 |
| | 15-30 | 851 | 0.68 | 6.83 | 0.01 | 0.10 | 11.83 | 0.61 | 0.60 | 7.40 | 2.28 | 0.09 | 12.78 | 3.95 |
| | 30-60 | 523 | 0.68 | 7.36 | 0.02 | 0.09 | 12.78 | 0.14 | 0.61 | 7.89 | 2.22 | 0.13 | 13.20 | 2.70 |
| | 60-100 | 675 | 0.68 | 7.79 | 0.02 | 0.07 | 13.43 | -0.28 | 0.61 | 8.33 | 2.21 | 0.12 | 13.65 | 1.97 |
| | 100-200 | 230 | 0.63 | 7.96 | 0.03 | 0.06 | 13.00 | 0.86 | 0.55 | 8.67 | 2.74 | 0.12 | 13.06 | 0.91 |
| BD | 0-5 | 153 | 0.62 | 0.12 | 0.00 | 0.12 | 0.20 | 0.01 | 0.53 | 0.13 | 0.01 | 0.02 | 0.27 | 0.15 |
| | 5-15 | 155 | 0.63 | 0.11 | 0.00 | 0.15 | 0.19 | 0.01 | 0.57 | 0.12 | 0.01 | 0.01 | 0.29 | 0.18 |
| | 15-30 | 155 | 0.60 | 0.11 | -0.00 | 0.11 | 0.19 | 0.01 | 0.54 | 0.13 | 0.01 | 0.01 | 0.27 | 0.12 |
| | 30-60 | 136 | 0.55 | 0.12 | -0.00 | 0.10 | 0.19 | -0.01 | 0.53 | 0.13 | -0.00 | 0.01 | 0.24 | 0.10 |
| | 60-100 | 95 | 0.57 | 0.12 | -0.00 | 0.10 | 0.19 | -0.01 | 0.51 | 0.13 | -0.01 | 0.02 | 0.24 | 0.07 |
| | 100-200 | 33 | 0.47 | 0.13 | 0.00 | 0.05 | 0.22 | 0.02 | 0.42 | 0.13 | -0.01 | 0.02 | 0.24 | 0.07 |

905

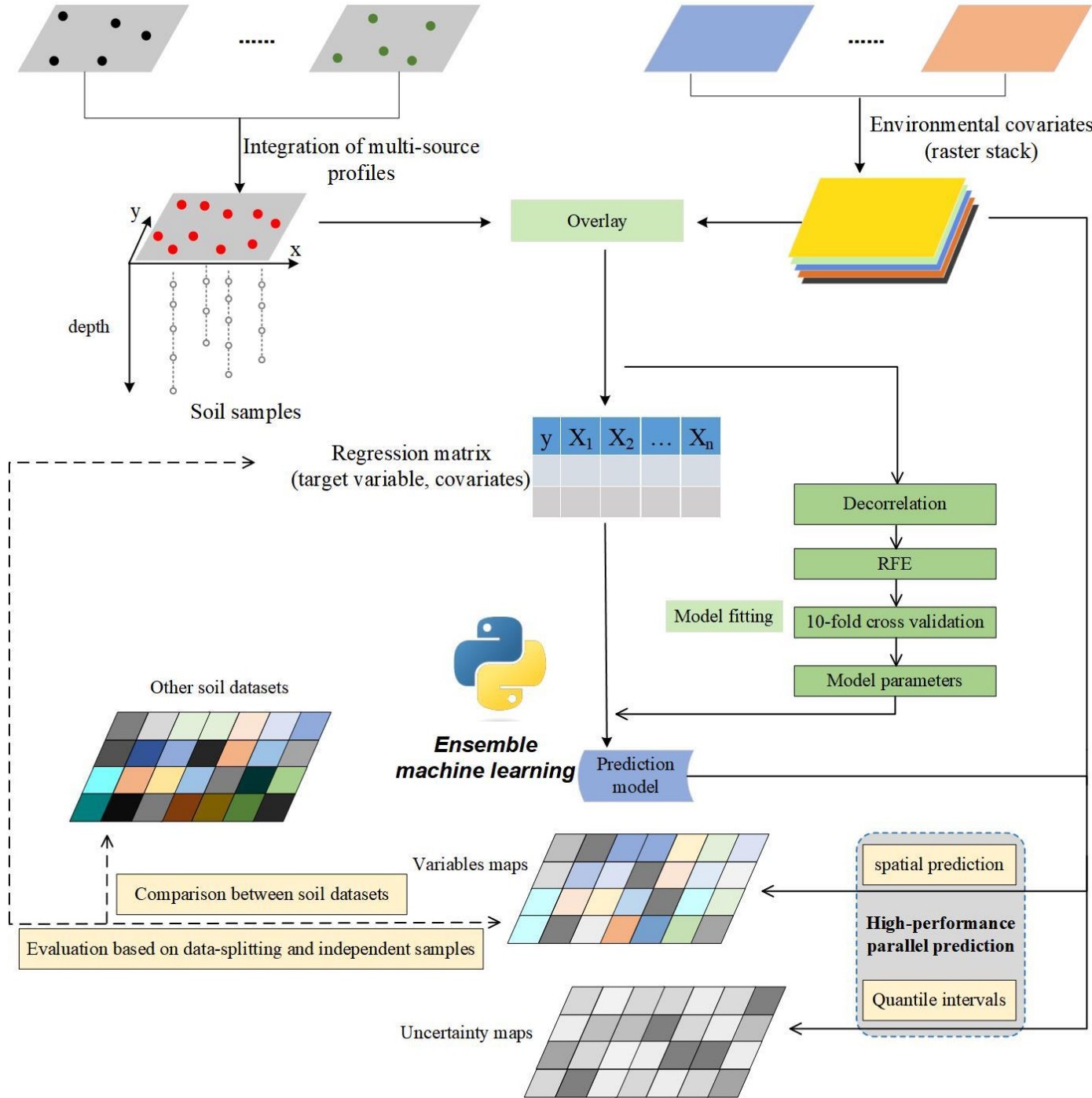

**Figure 1. The statistical framework for developing national-scale soil properties mapping in this study.**

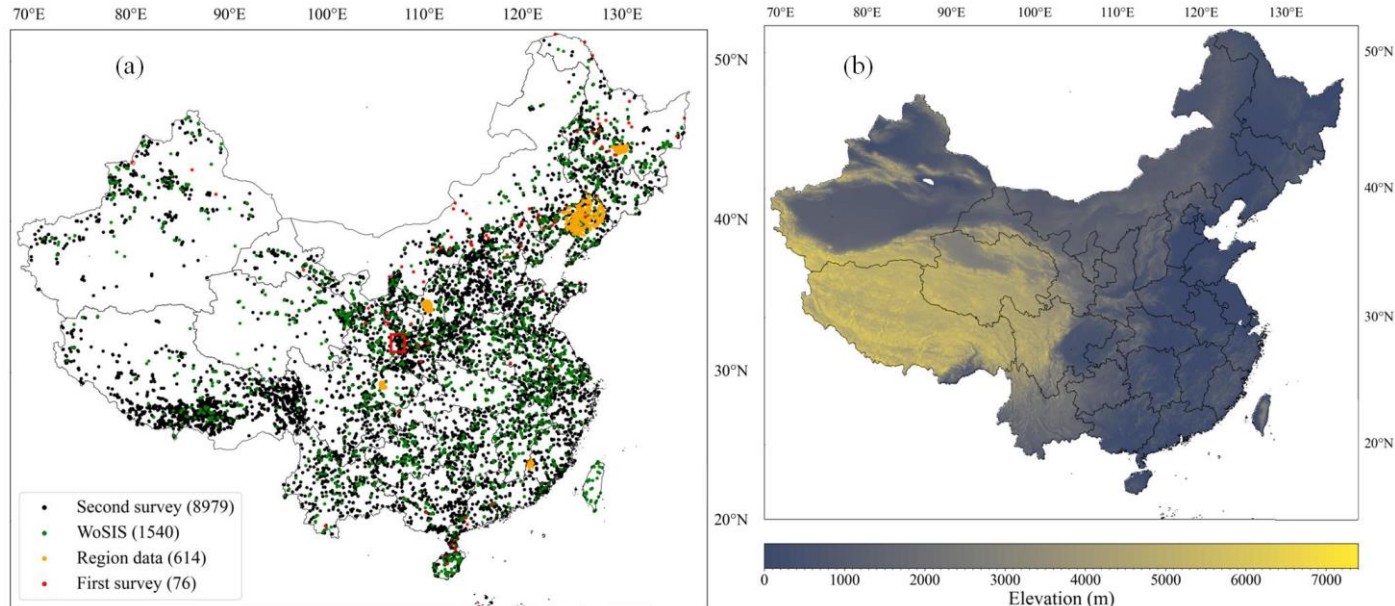

Figure 2. (a) Spatial distribution of the 11 209 soil profiles collected from various data sources: black dots indicate the Second National Soil Survey of China (Second survey), green dots correspond to World Soil Information Service (WoSIS), orange dots denote regional data, and red dots represent the First National Soil Survey of China (First survey). The red window indicates the area selected for visualizing the spatial patterns of soil properties. (b) Geographical map of the land area of China.

915

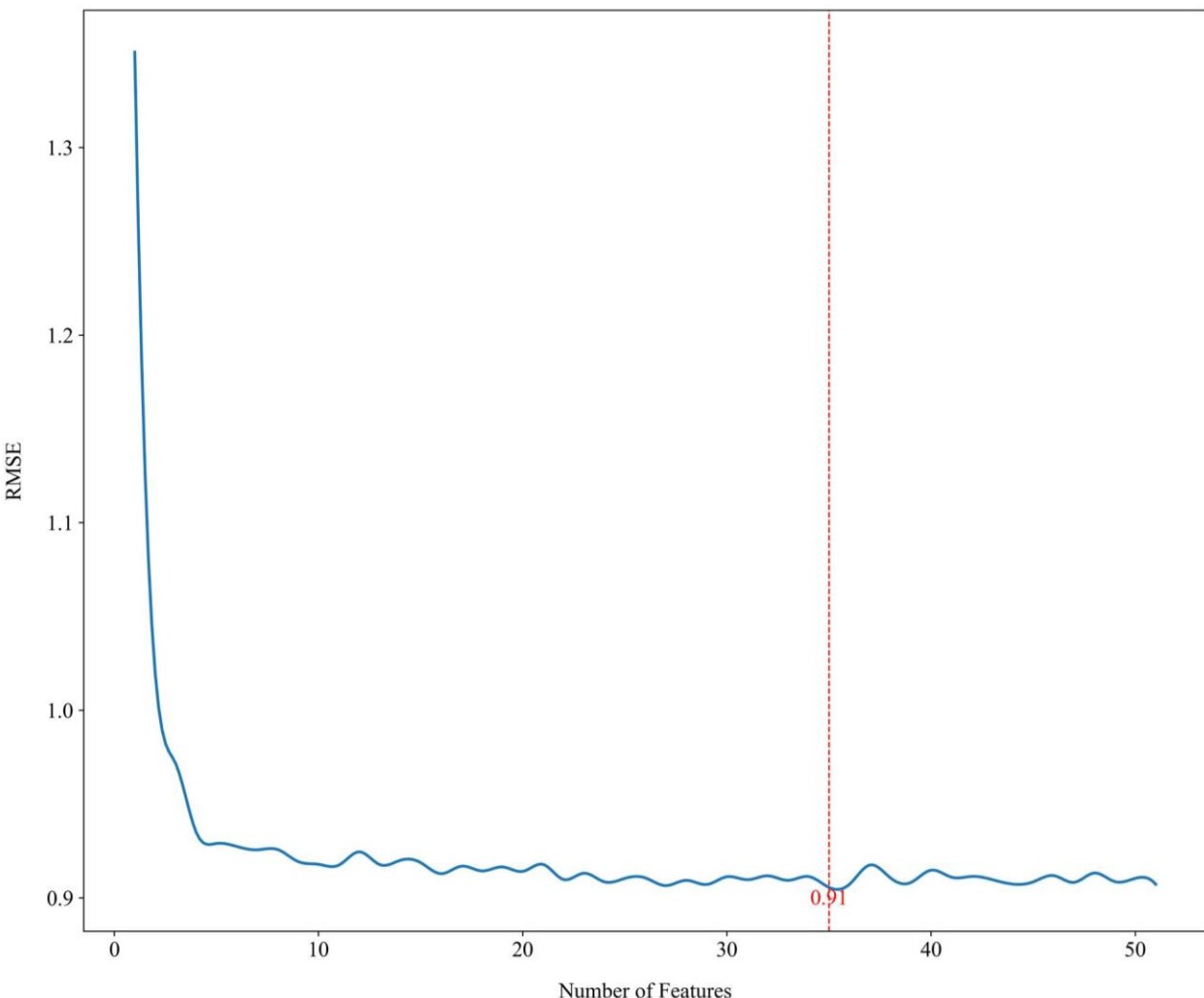

**Figure 3. Example of the loss function (RMSE) used in the Recursive Feature Elimination (RFE) step of covariates' selection for surface (0-5 cm) soil organic carbon content.**

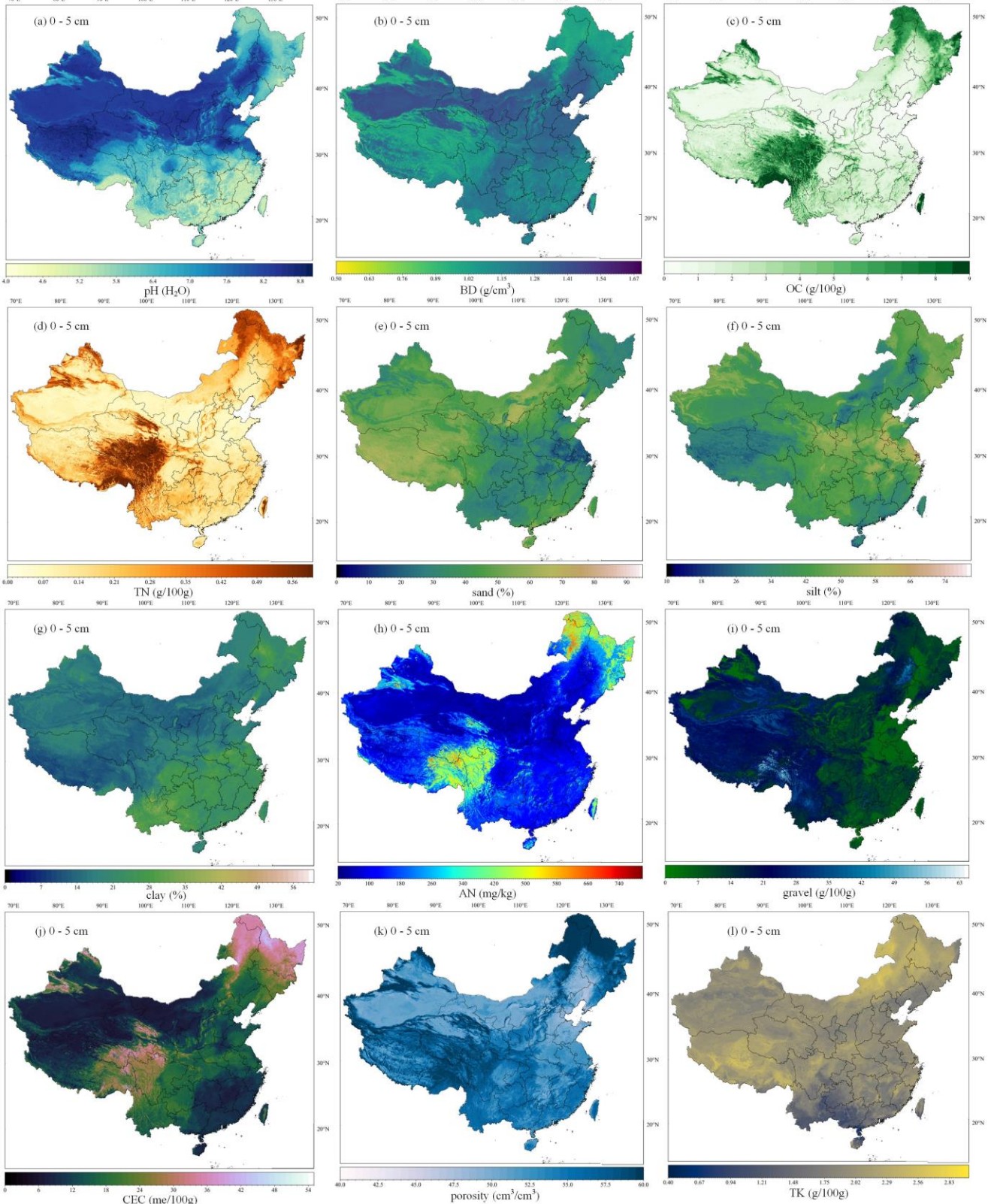

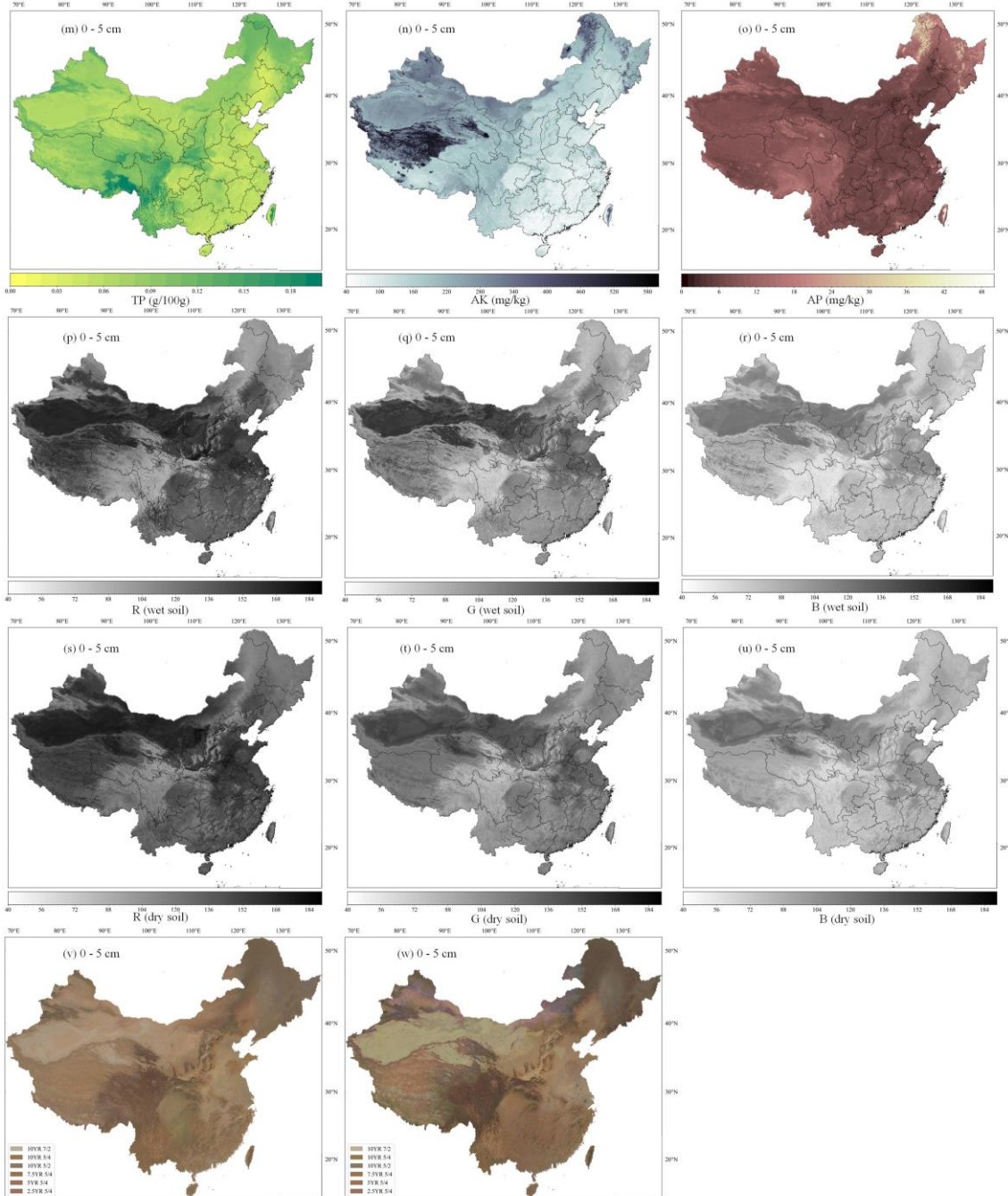

**Figure 4. The predicted maps of soil properties considered at the 0-5 cm depth interval for the land area of China. (a) pH (H₂O); (b) bulk density (BD); (c) soil organic carbon content (OC); (d) total nitrogen content (TN); (e,f,g) soil texture(sand, silt ,clay content); (h) alkali-hydrolysable nitrogen content (AN); (i) rock fragment content (gravel); (j) cation exchange capacity content (CEC); (k) porosity; (l) total potassium content (TK); (m) total phosphorus content (TP); (n) available potassium content (AK); (o) available phosphorous content (AP); (p,q,r) wet color (R, G, B); (s,t,u) dry color (R, G, B). (v) and (w) represent the dry and wet colors in the Munsell color system, respectively. See Figures S2-S24 in the appendix for the predicted maps of soil properties at all depth intervals.**

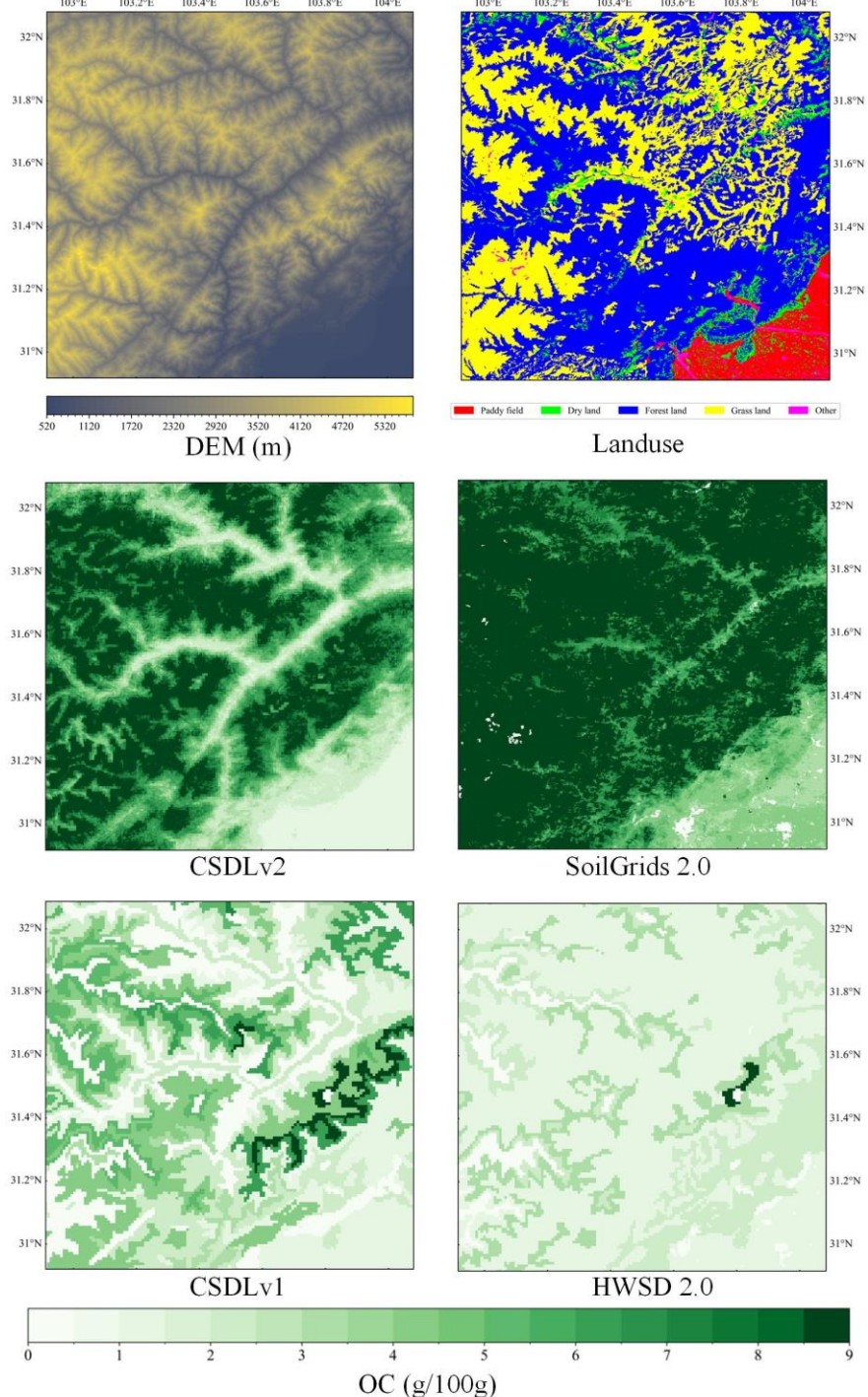

**Figure 5. Surface layer (0-5cm) soil organic carbon (OC) maps derived from our predictions (CSDLv2), SoilGrids 2.0, CSDLv1, and HWSD 2.0, respectively, in a selected area (102.92°-104.08°E and 30.92°-32.08°N) located in Sichuan Province. This selected area corresponds to the red window shown in Figure 1. DEM and landuse refer to the land surface elevation and land use type of the selected area, respectively.The spatial resolutions are 90 m for CSDLv2, 250 m for SoilGrids 2.0, and 1 km for both CSDLv1 and HWSD 2.0.**

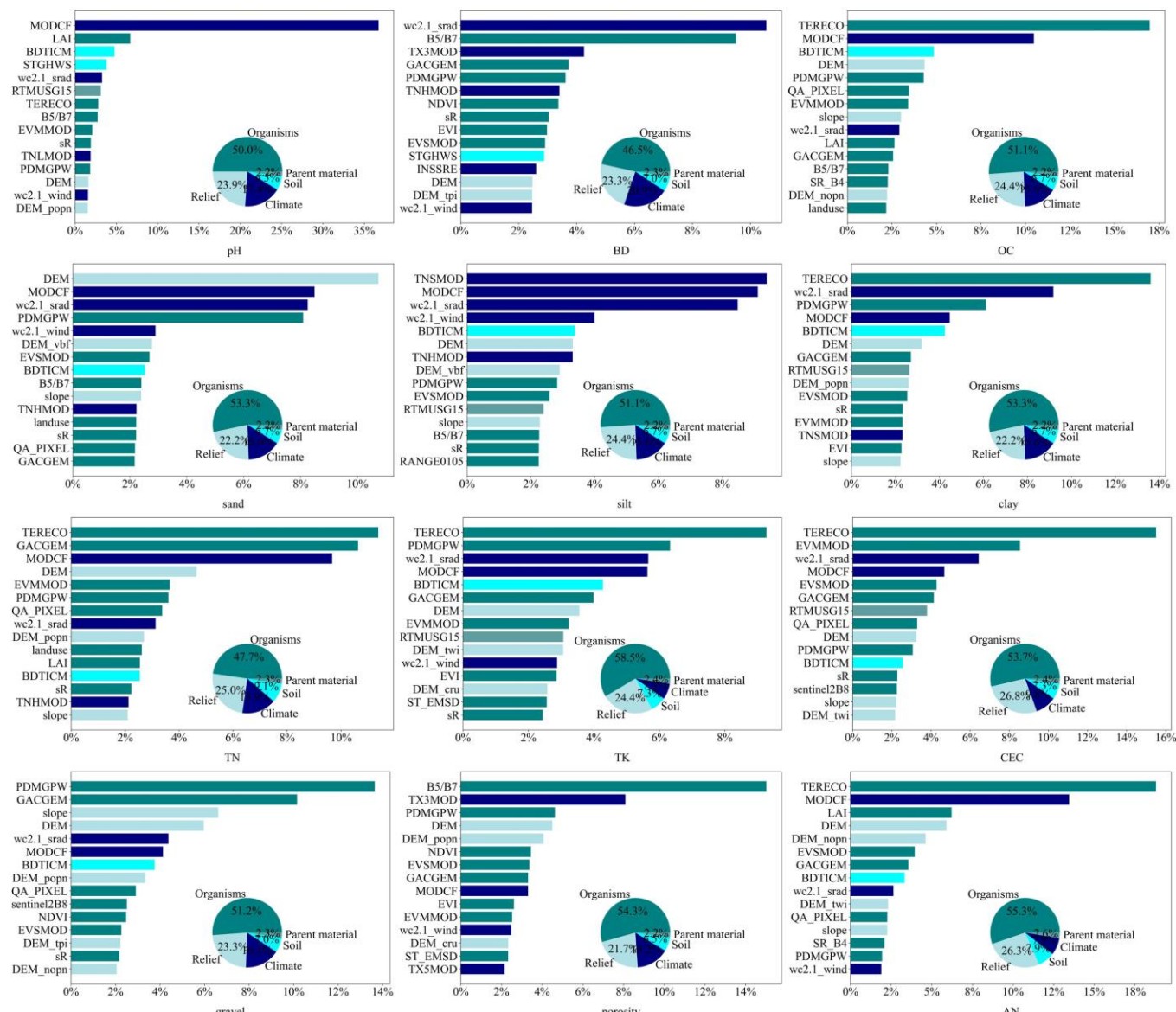

**Figure 6. Relative importance of the top 15 predictors for the Quantile Regression Forest model in the spatial predictions of soil pH, bulk density (BD), soil organic carbon (OC), soil texture (sand, silt, clay), total nitrogen (TN), total potassium (TK), cation exchange capacity (CEC), rock fragment (gravel), porosity, and alkali-hydrolysable nitrogen (AN) at the surface layer (0-5 cm). For other surface soil properties interested, including total phosphorus (TP), available potassium (AK), available phosphorus (AP), dry color (R, G, B), and wet color (R, G, B), see Figure S26. Refer to Table S1 in the appendix for abbreviations of the environmental covariates.**

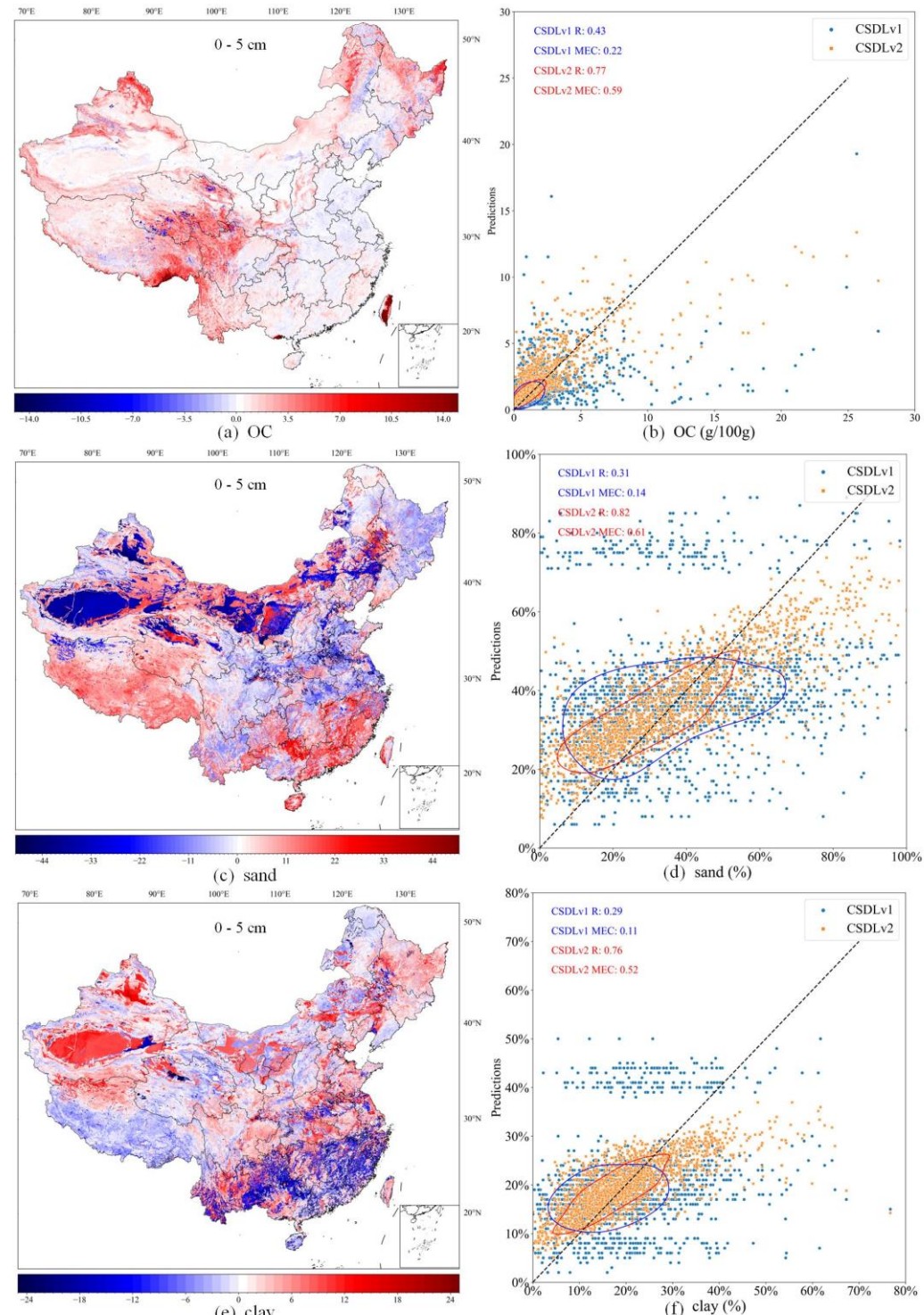

**Figure 7. Differences in predicted maps of soil organic carbon (a), sand (c), and clay (e) between CSDLv2 and CSDLv1 at the 0-5 cm depth interval and the corresponding scatter plots (b, d, f) indicating how well the predictions of CSDLv2 and CSDLv1 match the observations. The red and blue circles are bivariate kernel density estimates.**