# Peer review of "A China dataset of soil properties for land surface modeling (version 2, CSDLv2)"

_Earth System Science Data, 2024_

## Author Comment (AC1)

Dear Reviewer,

Thank you very much for your comments and professional advice. Your insights have significantly contributed to enhancing the academic rigor of our article. We appreciate the time and effort you devoted to reviewing our work. Based on your valuable suggestions and requests, we have implemented corrections and modifications to the revised manuscript. We believe these enhancements will further strengthen the quality of our work. We would like to provide a detailed account of the changes made:

Note: The modifications are shown in bold font. The comments are blue colored.

**GENERAL COMMENTS:**

**Comment #1: It would be informative to include a description of how the sand, silt, and clay content was computed. Did you consider using additive log-ratio transformed clay, silt, and sand content as the dependent variables to ensure that the sum of the three particle size fractions is 100%? Additionally, how was uncertainty computed for sand, silt, and clay content?.**
**Response:** Thank you for your insightful comments.

The soil particle size distribution in this study follows the International Society of Soil Science (ISSS) and Katschinski's schemes [Katschinski, 1956]. However, most land surface models (LSMs) require soil texture data in the FAO-USDA (United States Department of Agriculture) system. Therefore, the original ISSS and Katschinski particle-size distribution data were converted to the FAO-USDA system using several particle-size distribution models (Shangguan, 2013).

In this study, we did not use additive log-ratio transformations. However, we ensured that the sum of sand, silt, and clay content equaled 100%. The approach we used is as follows:

Initially, when we obtained the soil profile samples for sand, silt, and clay, we set a 5% threshold to filter out poor-quality samples. Specifically, samples where the sum of the three particle size fractions was greater than 105% or less than 95% were excluded (Shangguan et al., 2013).

After this step, we ensured data quality by controlling for sample integrity. We then built separate spatial prediction models for the three particle size fractions (sand, silt, and clay). After generating the spatial prediction maps for each fraction, we applied a weighting approach to ensure that the sum of the three fractions equaled 100%. We did not use additive log-ratio transformed method because we found that the predicted sums are not far from 100% in almost all cases and different methods (simple weighting and log-ratio) may not have big difference.

Regarding your question on uncertainty, we would like to clarify the following:

If you are referring to the uncertainty caused by particle-size distribution conversion, we did not estimate this uncertainty. As mentioned earlier, we applied a 5% threshold to filter out samples with significant conversion errors, and only the highquality samples were used. Different particle-size distribution conversion methods exist, and we chose the method that best suited our dataset, without estimating the uncertainty introduced by the conversion process itself.

However, if you are referring to the uncertainty of the mapping results in this study, we did estimate the uncertainty for the generated maps.

**Modification:**

**"For the sand, silt, and clay content from the FNSSC and SNSSC, they were measured following the schemes of International Society of Soil Science (ISSS) and Katschinski (Katschinski et al., 1956). Since most land surface models (LSMs) and other applications require soil texture data in the FAO-USDA system, we used several particle-size distribution models (Shangguan et al., 2013) to convert the original ISSS and Katschinski particle-size distribution data into the FAO-USDA system. A 5% quality control threshold was applied, excluding soil profile samples where the sum of the three fractions fell outside the 95%-105% range (Shangguan et al., 2013), and they were converted to make sure the sum of them are 100% by using the weighting approach. For the mapping of each particle size fraction (sand, silt, and clay), separate spatial prediction models were developed, and the weighting approach was applied to ensure that the sum of the three fractions equaled 100%."**

Katschinski, N. A.: Die mechanische bodenanalyse und die klassifikation der bo¨den nach ihrer mechanischen zusammensetzung, Pari, B, 321–327, 1956.

Shangguan, W., Dai, Y., Liu, B., Zhu, A., Duan, Q., Wu, L., Ji, D., Ye, A., Yuan, H., Zhang, Q., Chen, D., Chen, M., Chu, J., Dou, Y., Guo, J., Li, H., Li, J., Liang, L., Liang, X., Liu, H., Liu, S., Miao, C., and Zhang, Y.: A China data set of soil properties for land surface modeling, J. Adv. Model. Earth Syst., 5, 212–224, https://doi.org/10.1002/jame.20026, 2013.

**Comment #2: Please clarify in the entire manuscript whether data-splitting refers to cross-validation. If yes, please use the latter terminology, as it is more widely used.**

**Response:** Thank you for your valuable comment. In the manuscript, **data-splitting** does not refer to cross-validation. Data-splitting refers to dividing the dataset into training and test sets, where the model is trained on the training set and then evaluated on the test set.

The definition in the manuscript in section 2.3.2 is as follows:

**"The first method involved randomly selecting 10% of the multi-source soil profiles as test samples, while the remaining 90% were used for training the model (i.e., data-splitting)."**

The definition in the manuscript in section 2.3.1 is as follows:

**"This approach involved randomly dividing the training dataset into ten folds. One-tenth of these sub-datasets was utilized as the validation sample, while the remaining sub-datasets were applied for training the QRF model."**

Cross-validation is used during the model training phase for hyperparameter tuning using the training samples, whereas data-splitting is applied after

hyperparameter tuning to evaluate the model's performance on an independent test set. The two methods serve different purposes in the study.

**Comment #3: It is not clear how soil data from different time periods were considered for the mapping. This issue needs to be addressed in the manuscript.**
**Response:** Thank you for your insightful comment. In this study, for soil properties that are more sensitive to temporal changes, such as pH, OC, CEC, TN, TP, TK, AN, AK, and AP, we only used data from the Second National Soil Survey (1980s). As such, we have included the following statement in the manuscript: "the maps of CSDLv2 majorly represent the status of soil in 1980s."

We have revised the manuscript as follows: the first part has been included in the Materials and Methods section (Section 2.1.2), and the latter part has been incorporated into the Discussion section (Section 4.3):
**Modification:**

**"For soil properties sensitive to temporal changes, such as soil pH, organic carbon (OC), cation exchange capacity (CEC), total nitrogen (TN), total phosphorus (TP), total potassium (TK), alkali-hydrolysable nitrogen (AN), available phosphorus (AP), and available potassium (AK), we used only soil profile data from the SNSSC. In contrast, for properties less sensitive to temporal changes, such as sand, silt, clay, bulk density (BD), gravel, and porosity, we combined data from multiple sources. Since most soil profiles are from the SNSSC, the maps in CSDLv2 mainly represent the status of soil in the 1980s."**

**"Last but not least, although this study utilized multi-source soil profile data from different time periods to develop comprehensive static maps of soil properties, the CSDLv2 maps mainly represent the status of soil in the 1980s, as most soil profiles come from the SNSSC. For soil properties that change over time, other multi-source soil profile data have not been fully utilized."**

**Comment #4: For the description of the soil maps, it might be useful to add a geographical map of China and refer to physical features rather than only using compass directions. Please provide information on the spatial patterns (paragraph 3.3) of all the derived maps: maps of TN, AN, porosity, gravel, AP, and colours are not described.**
**Response:** Thank you for your valuable comment. In response to your suggestion regarding the description of the soil maps, we have added a geographical map of China showing the topography as a reference (**Figure 2**).

Additionally, as you recommended, **we have expanded the manuscript to include spatial pattern information for soil properties (TN, gravel, porosity AN, and AP)**, in addition to organic carbon (Figure 4). The spatial patterns of derived maps are now described in paragraph 3.3 and visualized in Figures S2-S26. In addition, we also added maps with spatial details in Fig.S26-30.
**Modification:**

**"Fig. S26-30 shows the spatial details of other soil properties, including TN, gravel, porosity AN, and AP."**

Since the soil color variation is not significant within small areas, we did not include its visualization in the manuscript.

**Comment #5: Data accessibility is problematic. I see that the maps could be downloaded through FTP, but it didn't work for me. The download site needs improvement, or information on how to use it should be provided.**

**Response:** Thank you for your valuable feedback. The data is hosted on a national data platform of China, and unfortunately, we do not have the authority to modify the download methods provided by the platform. Currently, File Transfer Protocol (FTP) is offered as the primary method for accessing large datasets, as it is generally more efficient and reliable than HTML downloads for handling large files. We recommend using common FTP clients like Filezilla (https://filezilla-project.org/) or FlashFXP (https://www.flashfxp.com/), which are widely used and support FTP downloads.

To further clarify the process, we have added a detailed explanation in the manuscript on how to download the dataset provided in this study:

**"Users can efficiently download the data using the File Transfer Protocol (FTP) account information provided at the above links and common FTP client tools such as Filezilla (https://filezilla-project.org/) or FlashFXP (https://www.flashfxp.com/)."**

**SPECIFIC COMMENTS:**

**Comment #6: TITLE: you could put into brackets the acronym of the database: (version 2)**

**Response:** Thank you for your suggestion. I have updated the title to include the acronym of the dataset as follows: "**A China dataset of soil properties for land surface modeling (version 2, CSDLv2).**"

**Comment #7: L21: please add information on accuracy based on all depth intervals, not only 0-5 cm.**

**Response:** Thank you for your valuable feedback. I have revised the manuscript to include accuracy information based on all depth intervals, as you suggested. The updated sentence now reads: "**The prediction accuracy of soil properties at all depth intervals ranged from good to moderate, with Model Efficiency Coefficients for most soil properties median ranging from 0.29 to 0.70 during data-splitting validation and from 0.25 to 0.84 during independent sample validation.**"

**Comment #8: L36: it seems to be Lu et al. (2016) based on reference list. Please check and correct.**

**Response:** Thank you for your careful review. After checking the reference list, we confirm that the citation is correct as Luo et al., 2016, not Lu et al., 2016. The reference is as follows:

Luo, Y., Ahlström, A., Allison, S. D., Batjes, N. H., Brovkin, V., Carvalhais, N., Chappell, A., Ciais, P., Davidson, E. A., Finzi, A., Georgiou, K., Guenet, B., Hararuk, O., Harden, J. W., He, Y., Hopkins, F., Jiang, L., Koven, C., Jackson, R. B., Jones, C. D., Lara, M. J., Liang, J., McGuire, A. D., Parton, W., Peng, C., Randerson, J. T., Salazar, A., Sierra, C. A., Smith, M. J., Tian, H., Todd-Brown, K. E. O., Torn, M., Van Groenigen, K. J., Wang, Y. P., West, T. O., Wei, Y., Wieder, W. R., Xia, J., Xu, X., Xu, X., and Zhou, T.: Toward more realistic projections of soil carbon dynamics by Earth system models, Global Biogeochemical Cycles, 30, 40–56, https://doi.org/10.1002/2015GB005239, 2016.

**Comment #9: L39: please cite other papers as well.**
**Response:** Thank you for your suggestion. I have now included additional citations, to provide a broader context. The updated sentence reads:
"**There is an urgent need for detailed, accurate, and up-to-date soil information to develop solutions for these challenges and to inform decision-making related to natural resource management (Arrouays et al., 2014; Dai et al., 2019b; Li et al., 2024).**"

Arrouays, D., Grundy, M. G., Hartemink, A. E., Hempel, J. W., Heuvelink, G. B. M., Hong, S. Y., Lagacherie, P., Lelyk, G., McBratney, A. B., McKenzie, N. J., Mendonca-Santos, M. d.L., Minasny, B., Montanarella, L., Odeh, I. O. A., Sanchez, P. A., Thompson, J. A., and Zhang, G.-L.: GlobalSoilMap:Toward a Fine-Resolution Global Grid of Soil Properties, in: Advances in Agronomy, vol. 125, Elsevier, 93–134, https://doi.org/10.1016/B978-0-12-800137-0.00003-0, 2014.

Dai, Y., Shangguan, W., Wei, N., Xin, Q., Yuan, H., Zhang, S., Liu, S., Lu, X., Wang, D., and Yan, F.: A review of the global soil property maps for Earth system models, SOIL, 5, 137–158, https://doi.org/10.5194/soil-5-137-2019, 2019b.

Li, T., Cui, L., Kuhnert, M., McLaren, T. I., Pandey, R., Liu, H., Wang, W., Xu, Z., Xia, A., Dalal, R. C., and Dang, Y. P.: A comprehensive review of soil organic carbon estimates: Integrating remote sensing and machine learning technologies, Journal of Soils and Sediments, https://doi.org/10.1007/s11368-024-03913-8, 2024.

**Comment #10: L47: please rephrase the following:** "exemplified by Brazil's", **the sentence is not finished.**
**Response:** Thank you for your feedback. I have revised the sentence as follows:
"**Additionally, broader-scale resolution maps, ranging from 250 to 5000 m, have also been investigated at the national level, exemplified by Brazil (Gomes et al., 2019).**"

**Comment # 11: L64:** ··· **(McBratney et al., 2003)** ··· **the mistyping errors of references could be prevented by using a referencing tool. Please recheck in the entire text if reference list is in line with their citations.**

**Response:** Thank you for your valuable feedback. I have carefully rechecked the entire manuscript and ensured that all references are now correctly aligned with their citations in the text. Any mistyping errors found have been corrected.

**Comment # 12: L68: please shortly describe in the text the limitations of the existing dataset.**

**Response:** Thank you for your valuable feedback. In response to your suggestion, we have now provided a concise description of the limitations of the existing dataset, which include the reliance on the traditional polygon linkage method, limited soil profile samples, and the lack of comprehensive soil property variables. These limitations have highlighted the necessity for a new version of the dataset to overcome these challenges.

**Modification:**

"**In summary, the existing dataset has several limitations, including its reliance on the traditional polygon linkage method, a limited number of soil profile samples, and the fact that it only contains basic soil property variables, lacking more comprehensive soil properties. Given these limitations, there is a compelling need to develop a new version of the dataset to address these challenges.**"

**Comment # 13: L80: please rephrase the following:** "soil specie survey", **it is mistyped.**

**Response:** Thank you for pointing that out. I have corrected the typo, and "soil specie survey" has been changed to "**soil series survey.**"

**Comment # 14: L83: please add the list of mapped soil properties.**

**Response:** I have updated the manuscript to include the list of mapped soil properties. The revised sentence now reads:

"**However, the study relied solely on a constrained set of about 4,500 soil profiles collected during the recent national soil survey, generating national grid maps for only some fundamental soil properties, including pH (H2O), organic carbon, cation exchange capacity, total nitrogen, total phosphorus, total potassium, bulk density, gravel content , soil texture, and soil thickness.**"

**Comment # 15: L84-85: please write with lower case letters the words** "available" **and** "alkali". **Is there a more general name for AN? E.g.: potential long-term supply of nitrogen in the soil (alkali-hydrolysable nitrogen, AN), or something similar?**

**Response:** Thank you for your helpful suggestions. I have revised the text to use lower case letters for "available" and "alkali," and I have also provided a more general description for AN as "**an index of the potential capacity of the soil to supply nitrogen.**" The revised sentence now reads:

"**The limitations stem from the absence of more comprehensive national grid maps for soil properties, including the fractions of total phosphorus and**

potassium readily available for plant absorption (available phosphorus, AP; available potassium, AK), an index of the potential capacity of the soil to supply nitrogen (alkali-hydrolysable nitrogen, AN), porosity, and others, imposing constraints on applications that necessitate a broader spectrum of soil properties information."

**Comment # 16: L94-107: please decrease repetition, by mentioning each advancements once in a logical order.**

**Response:** Thank you for your valuable feedback. I have reduced repetition by mentioning each advancement only once and reorganizing the points in a logical order.

**Comment # 17: L102: highlight that covariates were considered for the mapping as independent variables/predictors in the ML. Please consider that improvement in resolution is the result of points 1-3, therefore it could be mentioned after the points 1-4.**

**Response:** Thank you for your valuable feedback. I have highlighted that high-resolution environmental covariates were considered as predictors in the machine learning models, which contributed to the improvement in resolution. Additionally, the improvement in spatial resolution is now mentioned as the result of points 1-3, as suggested.

**Comment # 18: L104: please rephrase the following, it is difficult to understand:** "**without explicitly uncertainty estimates in CSDLv1**"

**Response:** Thank you for your feedback. I have rephrased the sentence for clarity. The revised version now explains that Quantile Regression Forests (QRF) were used in CSDLv2 to quantify prediction uncertainty, replacing the quality control information provided in CSDLv1, which did not include explicit uncertainty estimates. Additionally, point 4 (regarding uncertainty) has been removed in the revised manuscript following a logical restructuring.

**Comment # 19: L109: ⋯ in Fig. 2. ⋯ or change the order of Fig. 1 and 2.**

**Response:** We appreciate your feedback, and we have changed the order of Figures 1 and 2 accordingly.

**Comment # 20: L114-115: based on the entire manuscript 1) validation was performed based on data-splitting and independent soil profile dataset with measured soil data, and 2) comparison was done with existing national and global soil maps. Please consider it and revise the text and workflow figure (Fig. 2. left bottom corner) accordingly.**

**Response:** Thank you for your valuable feedback. I have revised the text and updated the workflow figure (**Fig. 2, left bottom corner**) accordingly to reflect the validation performed based on data-splitting and an independent soil profile dataset with measured soil data, as well as the comparison with existing national and global soil maps.

**Comment # 21: L132: ⋯ in Fig. 2 ⋯ change order of figures as suggested above.**

**Response:** Thank you for your suggestion. I have revised the text to reflect the change in the order of the figures, specifically updating it to "⋯ **in Fig. 2** ⋯" as recommended.

**Comment # 22: L150: it might be better to write "location" instead of "space".**

**Response:** I have updated the text to replace "space" with "**location**" as recommended.

**Comment # 23: L153: it is OK to use soil type information from HWSD, but please shortly explain why you used this 1 km resolution map instead of SoilGrids 250 m resolution.**

**Response:** Thank you for your insightful comment. Unfortunately, when we initially prepared the covariates, we did not fully consider the availability of the SoilGrids soil type map. Theoretically, both SoilGrids (250 m resolution) and HWSD (1 km resolution) soil type maps could be used in our study. However, it is difficult to definitively say which dataset is superior, as they are generated using different methods. Previous research (Chen et al., 2019; Wadoux et al., 2020) has shown that soil type plays a relatively minor role in large-scale soil property mapping, and its influence on the final results is usually limited. Nonetheless, we agree that using the SoilGrids soil type map could be valuable, and we will consider incorporating it as a covariate in future studies.

Chen, S., Liang, Z., Webster, R., Zhang, G., Zhou, Y., Teng, H., Hu, B., Arrouays, D., and Shi, Z.: A high-resolution map of soil pH in China made by hybrid modelling of sparse soil data and environmental covariates and its implications for pollution, Science of The Total Environment, 655, 273–283, https://doi.org/10.1016/j.scitotenv.2018.11.230, 2019.

Wadoux, A.M.J.-C., Minasny, B., McBratney, A.B., 2020. Machine learning for digital soil mapping: Applications, challenges and suggested solutions. Earth-Science Reviews 210, 103359. https://doi.org/10.1016/j.earscirev.2020.103359

**Comment # 24: L159: aspect is not included in Table S2, please add it or delete in the text if it was not used.**

**Response:** I have removed the reference to "**aspect**" from the text.

**Comment # 25: L160: do you mean "organism related covariate"? Please rephrase.**

**Response:** Thank you for your suggestion. I have revised the text to use "organism related covariates"

**Comment # 26: L174: similar as above, why soil factors were derived from HWSD 1 km, why not from SoilGrids 250 m?**

**Response:** Thank you for your comment. Please refer to the response to Comment # 23.

**Comment # 27: L181-182: please note that Pearson correlation coefficient can detect only linear relationships. Why didn't you let RFE decrease the number of covariates? Why did you consider first Pearson corr. coeff. to decrease the number of predictors?**

**Response:** Thank you for your valuable question. As shown in Figure 1, in this study, we first applied the Pearson correlation coefficient followed by Recursive Feature Elimination (RFE). The Pearson correlation coefficient offers a simple and efficient way to quickly identify linearly correlated redundant features, thereby reducing the number of covariates and the computational load for the subsequent feature elimination process.

After this initial filtering, RFE is applied to the reduced feature set for a more thorough evaluation, identifying the most important predictors that contribute to the model. This combination of methods allows us to leverage the strengths of both approaches: Pearson correlation is used to eliminate linear redundancy, and RFE is employed to refine feature selection, ultimately enhancing the model's efficiency and performance.

Furthermore, similar approaches have been adopted in other studies, where Pearson correlation was used first to filter linearly correlated features, followed by RFE for feature selection (Liu et al. 2022a; Poggio et al., 2021).

Liu, Wu, H., Zhao, Y., Li, D., Yang, J.-L., Song, X., Shi, Z., Zhu, A.-X., and Zhang, G.-L.: Mapping high resolution National Soil Information Grids of China, Science Bulletin, 67, 328–340, https://doi.org/10.1016/j.scib.2021.10.013, 2022a.

Poggio, L., De Sousa, L. M., Batjes, N. H., Heuvelink, G. B. M., Kempen, B., Ribeiro, E., and Rossiter, D.: SoilGrids 2.0: producing soil information for the globe with quantified spatial uncertainty, SOIL, 7, 217–240, https://doi.org/10.5194/soil-7-217-2021, 2021.

**Comment # 28: L201: could you please add in the supplementary material info about the 15 most important variables for all depth and soil properties? Similarly to Fig. S26, which shows it for depth 0-5 cm.**

**Response:** Thank you for your suggestion. We have added information about the 15 most important variables for all depths and soil properties in **Figures S31-S52**. We have added the following text to the manuscript:

**"The relative importance of the top 15 environmental covariates for other soil properties across all depths is visualized in Figures S31-S52."**

**Comment # 29: L201: please: mention somewhere under "2 Materials and Methods" how resolution of the derived maps was defined.**

**Response:** Thank you for your insightful comment. We chose to develop the soil property dataset at a 90-meter resolution because most of the high-resolution covariates used in our study are available at either 30-meter or 90-meter resolutions.

Importantly, several key environmental covariates, such as topography, are available at a 90-meter resolution, and considering the significant computational demands of generating a 30-meter resolution dataset, we determined that 90 meters would be the most suitable resolution for this study.

**Modification:**

**"The resolution of the derived soil property maps was defined based on the resolution of the available environmental covariates. Most high-resolution covariates are available at 30-meter or 90-meter resolutions, with key covariates like topography available at 90 meters. Considering both the availability of covariates and the computational cost of generating a 30-meter resolution dataset, we selected a 90-meter resolution as the target resolution for developing the soil property dataset in this study."**

**Comment # 30: L252: do you mean 1° × 1° tiles?**

**Response:** Yes, I have modified the corresponding content in the text to "**1° × 1° tiles.**"

**Comment # 31: L262: Please specify "four different values". Do you mean four prediction related values?**

**Response:** Thank you for your comment. We have revised the text as follows to enhance clarity:

**"Using the selected environmental covariates from the aforementioned feature engineering, the constructed model was applied to compute four statistical values—mean, 0.05 quantile ($q_{0.05}$), median (0.50 quantile, $q_{0.50}$), and 0.95 quantile ($q_{0.95}$)—at every 90 m pixel across all standard depth layers."**

**Comment # 32: L271-276: as mentioned above, please clarify if 10 fold cross-validation was performed. Does I mean that all the 1540 Chinese soil profiles of the WoSIS dataset was used only for validation?**

**Response:** Thank you for your comment. Please refer to the response to Comment # 2. Yes, all the 1540 Chinese soil profiles of the WoSIS dataset were used as test samples based on independent validation.

**Comment # 33: L314-315: please consider the following and rephrase if you agree: the goal might be to have training data that is representative for China's soil types. Do you think that the datasets available to train the model represents well the soil types under different land cover? I ask it because in the case of many countries soils from arable land are well represented, but soils from forested areas, or organic soils, or less widespread soils types are underrepresented. How it is in your case?**

**Response:** Thank you for your valuable comment. The phrase "**enhance the representativeness of the soil profile samples**" in the original sentence was not referring to an increase in the representativeness of specific soil types, but rather to a general increase in the spatial coverage of soil profile samples. By adding more soil profile samples, the overall representativeness increases, both in terms of spatial distribution and, to some extent, soil types. However, this does not fully address the issue of sample imbalance between different land cover types, such as the overrepresentation of samples from arable land and underrepresentation from forested areas or organic soils. Soil surveys have historically focused more on agricultural lands, and achieving a balance across different soil types and land covers remains a challenge.

We have modified the original text as follows:

"**As observed in Fig. S1, the probability density distributions of soil properties from multiple sources exhibit a generally similar trend, with minor differences that increase the spatial representativeness of the soil profile samples, rather than representing specific soil types.**"

**Comment # 34: L319-322: please note that vertical change in soil properties depends on soil types. Several soil properties are addressed in this manuscript, therefore be specific and do not state that "the average concentrations of most soil property variables tend to decrease with increasing depth (e.g., OC, TN), showing positive skewness distributions." The last statement is confusing: "indicating no statistically significant differences between samples from different depths." Is it the case for OC ?**

**Response:** Thank you for your valuable feedback, and I apologize for the confusion. I have revised the original text as follows:

Revised text: "**The vertical changes in soil properties vary depending on the specific soil property and soil type. For example, the content of OC and TN generally decreases with increasing depth in most soil types, exhibiting positive skewness distributions. However, other properties, such as soil pH or BD, show different vertical patterns depending on soil composition and local conditions.**"

Regarding your question on the statistical significance, I would like to clarify that in our case, the p-value for OC between different depths is less than 0.05, indicating significant differences in OC across different depths.

The revised the original text as follows:

"**Levene's test between samples from different depths yielded p-values greater than 0.05 for soil properties, indicating no statistically significant differences between samples from different depths.**"

**Comment # 35: L336: please add what can be the reason for the vertical decline in the predictability of soil texture.**

**Response:** Thank you for your insightful comment. I have added an explanation regarding the reasons for the vertical decline in the predictability of soil texture.

The revised text now as follows:

**"This decline may be attributed to the fact that environmental covariates primarily reflect surface conditions, leading to reduced correlation with deeper soil properties. Additionally, the decrease in sample size at greater depths may also contribute to this trend. Similar observations have been noted in other related studies (Liu et al., 2020; Poggio et al., 2021)."**

Liu, F., Zhang, G.-L., Song, X., Li, D., Zhao, Y., Yang, J., Wu, H., and Yang, F.: High-resolution and three-dimensional mapping of soil texture of China, Geoderma, 361, 114061, https://doi.org/10.1016/j.geoderma.2019.114061, 2020.

Poggio, L., De Sousa, L. M., Batjes, N. H., Heuvelink, G. B. M., Kempen, B., Ribeiro, E., and Rossiter, D.: SoilGrids 2.0: producing soil information for the globe with quantified spatial uncertainty, SOIL, 7, 217–240, https://doi.org/10.5194/soil-7-217-2021, 2021.

**Comment # 36:  L339: please add information about the deeper layers, as well.**

**Response:** Thank you for your suggestion. I have updated the text to include information about the prediction accuracy for organic carbon (OC) across all depth layers, rather than only focusing on the surface layer. The revised statement now reads:

**"The prediction accuracy for OC was relatively high, with approximately 25% to 60% of the variation in OC across all depth layers explained in both data-splitting and independent validation methods."**

**Comment # 37: L350: what do you mean by "regional covariates"? Please rephrase.**

**Response:** Thank you for your question. I have revised the text replacing "regional covariates." The updated sentence now reads:

**"Conversely, the prediction accuracy for soil pH slightly increased with depth. This improvement may be partly due to the increased stability of soil pH in deeper layers over large areas, leading to more consistent relationships with environmental factors (Liu et al., 2020)."**

Liu, F., Zhang, G.-L., Song, X., Li, D., Zhao, Y., Yang, J., Wu, H., and Yang, F.: High-resolution and three-dimensional mapping of soil texture of China, Geoderma, 361, 114061, https://doi.org/10.1016/j.geoderma.2019.114061, 2020.

**Comment # 38: L356: Fig. 4 is discussed later than Fig. 5, please change order of the figures.**

**Response:** Thank you for your comment. I have rearranged the figures and changed the order of Figures 4 and 5 accordingly.

**Comment # 39: L357-358: please explain more the fact that you describe in sentence starting with "The gross …". Please note that values of soil properties not always increase with depth.**

**Response:** Thank you for your comment. We agree that not all soil properties increase with depth, and we have removed the original sentence for clarity.

**Comment # 40: L364: please rephrase the first sentence, it is not complete.**
**Response:** Thank you for your feedback. I have revised the first sentence to improve its completeness.
The updated sentence now reads:
     **"As shown in Fig. 4(b) for BD, northern regions tend to have higher bulk density due to low organic matter content and frequent agricultural activities."**

**Comment # 41: L366: please explain what you mean by "looser soil particles". What is the reason of having .lower bulk density in the Qianghai-Tibet Plateau?**
**Response:** Thank you for your question. By "looser soil particles," I meant higher porosity, which refers to soils with more pore space between particles, resulting in lower compaction. In regions with higher organic matter content, such as the southern areas, soils typically have a more porous structure, leading to lower bulk density (BD). Regarding the Qinghai-Tibet Plateau, the lower BD in this region is primarily attributed to the higher organic carbon (OC) content, which also results in lower compaction and density.
For clarity, we have revised the original sentence to:
     **"Southern regions generally have lower bulk density owing to higher organic matter content and higher porosity."**

**Comment # 42: L367: Is land use the only factor that influence BD in the south-eastern coastal areas? Please explain differences in BD in deeper horizons, which are less affected by land use.**
**Response:** Thank you for your insightful comment. To clarify, the original sentence refers specifically to surface soils, which are more affected by land use practices. we have revised the original sentence to avoid confusion as follows:
     **"Southeastern coastal areas show significant variation in surface bulk density, heavily influenced by land use practices."**
     Regarding the deeper soil horizons, we did not discuss them in this study. However, bulk density in deeper layers is influenced by a variety of factors such as geological conditions, parent material, and compaction from overlying soil layers. These factors play a more dominant role in deeper horizons compared to surface soils, which are primarily affected by land use.

**Comment # 43: L368: please rephrase the first sentence, it is not complete.**
**Response:** Thank you for your feedback. I have revised the first sentence to improve its completeness.
The updated sentence now reads:
     **"As shown in Fig. 4(c), OC content decreases from southeast to northwest, corresponding with the influence of the southeast monsoon."**

**Comment # 44: L368-373: please be more specific in referencing the specific regions. Present sentences are contradicting, due to specifying the locations based on the points of the compass.**

**Response:** Thank you for your comment. Please refer to the response to comment #4. The updated sentence now reads:

**"As shown in Fig. 4(c), the spatial predictions of OC content reveal significant regional differences. The highest OC levels are found in the eastern Tibetan Plateau, northeastern China, and northern Xinjiang, where human activities are minimal. In contrast, the lowest OC content is observed in the northwestern desert regions. OC content shows a decreasing trend from southeast to northwest, corresponding to the influence of the southeast monsoon. OC content is closely related to climatic conditions and land use practices (Zhang et al., 2023b; Zhou et al., 2019b). The spatial pattern of total nitrogen (TN) is similar to that of OC content."**

**Comment # 45: L373: please discuss map of TN, and why it shows similar pattern with OC.**

**Response:** Thank you for your comment. I have revised the text to explain why the spatial pattern of total nitrogen (TN) is similar to that of soil organic carbon (SOC). The revised sentence now as follows:

**"Areas with high precipitation and good vegetation cover tend to have higher OC and TN levels, while areas with low precipitation and poor vegetation cover tend to have lower OC and TN levels. This is because both OC and TN are closely related to organic matter input from vegetation. In regions with high vegetation productivity, organic matter contributes to both carbon and nitrogen accumulation in the soil, resulting in similar spatial patterns for OC and TN."**

The revised sentence now as highlights that both OC and TN are influenced by similar factors, such as precipitation and vegetation cover, which lead to their comparable spatial distributions. The accumulation of organic matter, driven by vegetation, contributes to the levels of both carbon and nitrogen in the soil, hence the similar patterns observed in the maps.

**Comment # 46: L400: ⋯ lists the PICP values ⋯**

**Response:** We have changed the text to "**...lists the PICP values ...**".

**Comment # 47: L410: please discuss how uncertainty changes with soil depth. What can be an explanation for that change?**

**Response:** Thank you for your valuable comment. We have added a discussion on how uncertainty changes with soil depth in the revised manuscript.
The revised sentence as follows:

**"As soil depth increases, the uncertainty in predictions for properties like OC and pH generally decreases due to the more stable nature of subsurface layers, reduced influence from external factors, and the fact that deeper soils are less affected by environmental covariates. Additionally, while topsoil is more**

complex and variable due to its interaction with the environment, subsurface layers tend to have more consistent properties, leading to a less uncertainty in predictions at depth (Liu et al., 2022a)."

The revised text now includes explanations for the observed changes in uncertainty, specifically relating to the influence of soil composition and environmental factors at different depths.

Liu, Wu, H., Zhao, Y., Li, D., Yang, J.-L., Song, X., Shi, Z., Zhu, A.-X., and Zhang, G.-L.: Mapping high resolution National Soil Information Grids of China, Science Bulletin, 67, 328–340, https://doi.org/10.1016/j.scib.2021.10.013, 2022a.

**Comment # 48: L413-414: do you mean that organism type variables have the highest variable importance? Please rephrase.**
**Response:** Yes, we have changed the text as follows:

"Overall, organism-type covariates account for a significant proportion among different categories of environmental factors."

**Comment # 49: L422: please note that soils developed on shallow bedrock do not always have low OC. Vegetation type on those soils influence the rate of OC accumulation.**
**Response:** Thank you for your insightful comment. We have revised the sentence for clarity as follows:

"Shallow bedrock typically results in thinner soil layers, which can limit soil development and the accumulation of OC. However, soils developed on shallow bedrock do not always have low OC, as the rate of OC accumulation can be significantly influenced by the type of vegetation present."

**Comment # 50: L437: what is the source of organic matter content (TERECO) input layer in the case of clay content maps of CSDLv2? Isn'ｔ it terrestrial ecosystems? Please revise the sentence.**
**Response:** We have revised the original sentence to:

"For clay prediction, organism-type covariates (e.g., TERECO, Table S1) rank as the most important environmental covariate."
The source of the TERECO input layer is provided in Table S1 (https://landscape12.arcgis.com/arcgis/rest/services/World_Terrestrial_Ecosystems/ImageServer).

**Comment # 51: L441－443: please rephrase the last two sentences of the paragraph, those are difficult to understand.**
**Response:** Thank you for your comment. The last two sentences did not add substantial meaning, so we have removed them for clarity.

**Comment # 52: L446-447: please provide more information about the results of Shrini at el. (2017). It is not clear how that is related to your results on CEC.**

**Response:** Thank you for your comment. We have added the following text to further explain the relevance of Shiri et al. (2017) to our findings:

**"Shiri et al. (2017) investigated the relationships between soil carbon content, clay content, and particle size with CEC. They found that higher organic carbon and clay content significantly enhance CEC due to their high specific surface areas and cation retention capacities. This is consistent with our findings, where areas with higher organic content, influenced by plant root activity, showed higher CEC value."**

Shiri, J., Keshavarzi, A., Kisi, O., Iturraran-Viveros, U., Bagherzadeh, A., Mousavi, R., and Karimi, S.: Modeling soil cation exchange capacity using soil parameters: Assessing the heuristic models, Computers and Electronics in Agriculture, 135, 242–251, https://doi.org/10.1016/j.compag.2017.02.016, 2017.

**Comment # 53: L451: ⋯ SoilGrids 2.0 ⋯ please correct it here and the entire manuscript.**

**Response:** We checked the full text, as well as the corresponding content modified to '... SoilGrids 2.0 ...'.

**Comment # 54: L452: please add the selection criteria both in the text and caption of Table 3. E.g., soil properties with highest prediction accuracy, or something similar.**

**Response:** Thank you for your comment. We have revised the manuscript and Table 3 accordingly. The text now reads:

**"Table 3 lists the validation accuracy of selected soil properties with the highest prediction accuracy using the data-splitting validation method."**

Additionally, the caption of Table 3 has been updated to:

**"Table 3. Accuracy evaluation of the selected soil properties with the highest prediction accuracy in CSDLv2, CSDLv1, SoilGrids 2.0, and HWSD 2.0, based on the randomly held-back soil profiles. ……"**

**Comment # 55: L458-462: In the case of MEC calculate a percentage improvement relative to the possible range or describe absolute improvement, e.g. MEC improved from 0.48 to 0.69.**

**Response:** Thank you for your suggestion. In response to your comment, we have revised the manuscript to describe the absolute improvement in MEC values instead of calculating a percentage improvement. The text now reflects this change, with examples of absolute improvements included.

**Modification:**

**"Specifically, using data-splitting validation as an example, our predictions for pH demonstrated an absolute improvement in the mean MEC for all layers, increasing from 0.60 to 0.70, while the RMSE decreased from 0.77 to 0.68 compared to SoilGrids 2.0. In comparison to CSDLv1, our prediction**

**performance for pH improved from 0.44 to 0.70, with the RMSE reducing from 0.96 to 0.68. ……"**

**Comment # 56: L477-479: do you think that CSDLv2 can better capture sites with extremely low or extremely high values? If yes, please add it and discuss why it can describe better the extreme values than the other maps.**

**Response:** Thank you for your insightful comment. Yes, CSDLv2 can better capture sites with extremely low or extremely high values. In response, we have added discussion in the manuscript:

**"Additionally, Figure 7 illustrates the ability of CSDLv2 and CSDLv1 to capture site test values, showing that CSDLv2 is more effective in capturing extreme values observed at the sites."**

**Comment # 57: L493-494: please add an example for "smoothing the properties of certain regions". Or rephrase the sentence. Do you mean that extreme values are smoothed due to the type of algorithm used (QRF – provides a mean of several trees, which includes a mean at each node)?**

**Response:** Thank you for your valuable comment. We have rephrased the sentence for clarity as follows:

**"This may be due to the better fitting ability of DSM technology with the available data. However, the use of the QRF algorithm, which averages predictions from multiple trees, tends to smooth out extreme values during spatial extrapolation, potentially reducing variability in certain regions."**

**Comment # 58: L496: ⋯ To show the impact of the ⋯ or something similar.**

**Response:** We made the following changes:

**"Further studies are needed to demonstrate the impact of the new soil dataset compared to the old version and global soil datasets by running a land surface model."**

**Comment # 59: L499: ⋯ aspects. ⋯ end the sentence, delete ":".**

**Response:** We have removed the ':'.

**Comment # 60: L499: is the resolution of the derived maps 90 m, because the input layers, which are most important for the predictions, also have this resolution? If yes, please add this shortly.**

**Response:** Thank you for your comment. Please refer to response #29. We have modified the text as follows:

**"First, CSDLv2's spatial resolution is 90 m, aligning with the resolution of the most important input layers used for the predictions, and this is an improvement over CSDLv1's 1 km resolution."**

**Comment # 61: L505-506: please add the benefit of producing map of soil colours in RGB. What is its practical use?**

**Response:** Thank you for your comment. We have modified the text as follows:

**"Third, an RGB soil color system (i.e., red, green, and blue) has been added, resolving the inconvenience of only having the Munsell color system in the first edition dataset. This addition enhances the visual representation of soil colors and allows for better integration with digital platforms, remote sensing applications, and computer displays (Al-Naji et al., 2021)."**

Al-Naji, A., Fakhri, A.B., Gharghan, S.K., Chahl, J., 2021. Soil color analysis based on a RGB camera and an artificial neural network towards smart irrigation: A pilot study. Heliyon 7, e06078. https://doi.org/10.1016/j.heliyon.2021.e06078

**Comment # 62: L523-524: the meaning of the sentence starting with "These soil nutrients …" is not clear, please describe more. Do you mean warm-up period?**

**Response:** Thank you for your comment. Yes, "spin-up" is a term commonly used in the modeling field, particularly among researchers working on land surface models, and it refers to the "warm-up period."

We have revised the text to clarify this, stating:

**"These soil nutrients can be calculated by running models for thousands of years until an equilibrium state is reached, a process known as model 'spin-up' (i.e., warm-up period)."**

**Comment # 63: L534: do you think that 90 m resolution can meet the needs of precision agriculture? 90 m resolution might support the spatial delineation of management zones. Please consider to revise it in the text.**

**Response:** Thank you for your valuable comment. We have made the following revision to the text:

**"as well as supporting the spatial delineation of management zones in precision agriculture."**

**Comment # 64: L557-558: "soil management" or "land use"? Please revise if land use is the correct word.**

**Response:** Thank you for pointing out the problem. I have changed 'soil management' to '**land use**'.

**Comment # 65: L557-571: this description is very informative. Suggestion for future development: if elevation and slope is highly correlated with temperature and precipitation, it might be possible to derive 90 m resolution climate variables from the original 1 km resolution – downscaling – based on topographical variables.**

**Response:** Thank you for your insightful comment. While downscaling is an interesting approach, it is not directly relevant to the scope of this study. There are already many studies that have used topography-based downscaling methods to produce high-resolution climate variables, such as temperature and precipitation (e.g., Chen et al., 2024). Downscaling is a well-established field with many mature methods.

As it falls outside the primary focus of this study, we did not incorporate it into our research. However, we added the following sentence expressing suggestion for future development:

**"Topography downscaling methods can be used to prepare high-resolution climate covariates (Chen et al., 2024)"**

Chen, S., Li, L., Dai, Y., Wei, Z., Wei, N., Zhang, Y., Zhang, Shupeng, Yuan, H., Shangguan, W., Zhang, Shulei, Li, Q., 2024. Exploring Topography Downscaling Methods for Hyper-Resolution Land Surface Modeling. Geophysical Research: Atmospheres [preprint]. https://doi.org/10.22541/au.171403656.68476353/v1

**Comment # 66: L572-578: Ok, but it is not clear how you handled soil data originating from different time periods in your study. Please explain it shortly in the text.**
**Response:** Thank you for your comment, please refer to response #3.

**Comment # 67: L583: on the download page why:**
**- temporal resolution is yearly and**
**- spatial resolution is 10 m – 100 m?**
**Response:** Thank you for your suggestion. We have added the following brief explanation in the text regarding how the 1 km and 10 km resolutions were derived:

**"To meet the spatial resolution requirements of different applications, CSDLv2 not only provides soil properties at a 90 m resolution but also offers at 1 km and 10 km resolutions. These 1 km and 10 km resolution data were derived from spatial predictions made by the constructed model using environmental covariates at the corresponding resolutions."**

The dataset developed in this study has a maximum resolution of 90 meters. However, on the data platform, we were only able to select a resolution range (without the ability to specify an exact number), so we chose the most appropriate range of 10-100 meters. Additionally, we set the temporal resolution to 'Null' to indicate that the data is static.

**Comment # 68: L589: ⋯ soil physical and chemical soil properties, with ⋯ Please delete here and in the entire manuscript the word "fertility". Fertility is a complex soil property defined by many indicators. In this manuscript soil physical and chemical properties were addressed Of course these influence soil fertility, but the focus is not on that in the manuscript.**
**Response:** Thank you for your comment. We have removed the word "fertility" from this section as well as from the entire manuscript, as you suggested.

**Comment # 69: L594: ⋯ gridded soil datasets, ⋯**
**Response:** We have revised the text to "...gridded soil datasets, ... "

**Comment # 70: L594: please rephrase "more reasonalble", with something more specific.**

Response: Thank you for your comment. I have revised the text to replace "more reasonable" with a more specific phrase.

The updated sentence now reads:

"**CSDLv2 provided more spatial details and better represented the spatial variation characteristics of soil properties in China compared to other soil products.**"

**Comment # 71: L596: please shortly indicate that CSDLv2 describes the state of 1980.**

Response: Thank you for your comment.

The updated sentence now reads:

"**Furthermore, as this dataset is primarily based on legacy soil profiles from the Second National Soil Survey of China and describes the state of soil properties in the 1980s, it serves as a valuable complement to maps based on 2010s soil profiles, providing new perspectives for studying temporal changes in soil properties.**"

**Comment # 72: L599-601: please complement the last sentence by how the limitations of CSDLv2 could be addressed in future studies – i.e., summarize paragraph 4.3.**

Response: Thank you for your valuable feedback. The limitations of CSDLv2 can be addressed in future studies by focusing on several areas of improvement: incorporating high-resolution remote sensing data, developing more accurate 3D models that account for vertical soil variability, and addressing the temporal changes in soil properties by using data from multiple time periods. These improvements will help refine soil property mapping and provide more accurate and dynamic soil information.

The updated sentence now reads in the conclusion:

"**Future work can improve soil property mapping by employing advanced deep learning methods and incorporating more observations, particularly in regions with sparse samples like western China. Additionally, integrating high-resolution remote sensing data, developing more accurate 3D models, and accounting for temporal changes in soil properties will further enhance the mapping accuracy and usefulness of CSDLv2.**"

DATA AND CODE AVAILABILITY:

**Comment # 73: The codes are accessible at GitHub.**

**Data accessibility is not smooth. I see that the data could be downloaded through FTP, but it didn't work for me. The download possibility needs improvement or information on using the download site is needed.**

Response: Thank you for your comment, please refer to response #3.

**Comment # 74: Table 2: is it possible to give a general variable name for "Sentinel2B2/B3/B4/8/9" under the description column?**

**Response:** Thank you for your suggestion. We have revised the description column by replacing "Sentinel2B2/B3/B4/8/9" with the more general term "**Sentinel-2 (B2, B3, B4, B8, B9)**". Similarly, "SR_B4/B5/B6/B7" has been updated to "**SR (B4, B5, B6, B7)**" for consistency.

**Comment # 75: Table 3: do you mean that it is the result of cross-validation in the case of CSDLv2 and performance of the other maps (CSDLv1, SoilGrids 2.0, HWSD 2.0) on the dataset used to train and test the CSDLv2 predictions? Please revise the title to increase clarity. Add number of samples considered for the validation in a separate column.**

**Response:** Thank you for your comment, and I apologize for the confusion. What we intended to convey is that 10% of the randomly held-back samples were used as the test set. We then evaluated the accuracy of the four maps (CSDLv2, CSDLv1, SoilGrids 2.0, HWSD 2.0) based on this test set.

The title in Table 3 has been revised as follows to increase clarity:

**"Accuracy evaluation of the selected soil properties with the highest prediction accuracy in CSDLv2, CSDLv1, SoilGrids 2.0, and HWSD 2.0, based on the randomly held-back soil profiles. The "Number" column indicates the number of samples used during testing. Refer to Table S4 for the complete accuracy evaluation of the soil properties considered. See Table 1 for the abbreviations and units of the soil properties of interest."**

**Comment # 76: Fig. 2: revise left bottom corner based on advice for L114-115, and reedit the figure of "Other soil datasets", its pattern might not be the same as that of the "Variable maps". Direction of arrow on the left might go from "points of the soil profiles" to "Compare and evaluate".**

**Response:** Thank you for your suggestion, we have revised it as you suggested.

**Comment # 77: Fig. 4: the caption does not include information on DEM and land use map. Please add them.**

**Response:** Thanks to your reminder, we have added the DEM and land use map information to the Fig. 5 (Fig. 4 in the pre-revision text) caption.

The full title of the revised Fig. 5 is:

**"Surface layer (0-5cm) soil organic carbon (OC) maps derived from our predictions (CSDLv2), SoilGrids 2.0, CSDLv1, and HWSD 2.0, respectively, in a selected area (102.92°-104.08°E and 30.92°-32.08°N) located in Sichuan Province. This selected area corresponds to the red window shown in Figure 1. DEM and landuse refer to the land surface elevation and land use type of the selected area, respectively.The spatial resolutions are 90 m for CSDLv2, 250 m for SoilGrids 2.0, and 1 km for both CSDLv1 and HWSD 2.0."**

**Comment # 78: Fig. 5: the labels are not visible. Please consider to show the maps in two or three figures, to increase visibility and readability. Please find a logic to put the maps into two or three groups, than you do not have to fit all 23 maps to one page (one figure), but to two or three figures. Please add unit of the soil properties and add "content" where needed, e.g.: sand, silt, and clay content, etc.**

**Response:** Thank you for your helpful suggestion. To increase visibility and readability, we have followed your recommendation and split the 23 maps into two figures (two pages). We have also added the units for the soil properties and included the word "content" where necessary.

The revised full caption for Fig. 4 is as follows:

**"The predicted maps of soil properties considered at the 0-5 cm depth interval. (a) pH (H2O); (b) bulk density (BD); (c) soil organic carbon content (OC); (d) total nitrogen content (TN); (e,f,g) soil texture(sand, silt ,clay content); (h) alkali-hydrolysable nitrogen content (AN); (i) rock fragment content (gravel); (j) cation exchange capacity content (CEC); (k) porosity; (l) total potassium content (TK); (m) total phosphorus content (TP); (n) available potassium content (AK); (o) available phosphorous content (AP); (p,q,r) wet color (R, G, B); (s,t,u) dry color (R, G, B). (v) and (w) represent the dry and wet colors in the Munsell color system, respectively. See Figures S2-S24 in the appendix for the predicted maps of soil properties at all depth intervals."**

**Comment # 79: Fig. 6: please increase size of the letters on the plot, it is difficult to read.**

**Response:** We have increased the size of the letters on the plot as you suggested.

**Comment # 80: Fig. 7: please:**
**- increase size of the letters on the plot, it is difficult to read,**
**- add R2 – for both maps – and 1:1 line to b), d) and f) plots to better see the comparison,**
**- use the same min and max values on x and y axis by soil properties, e.g.: 0 and 30 % for OC, 0 and 100 % for sand, 0 and 80 % for clay.**

**Response:** Thank you for your helpful suggestions. We have revised Figure 7 accordingly. The size of the letters on the plot has been increased for better readability. Additionally, we have added $R^2$ values for both maps and included the 1:1 line in the b), d), and f) plots to improve the comparison. The min and max values on the x and y axes have also been adjusted for each soil property (e.g., 0 to 30% for OC, 0 to 100% for sand, and 0 to 80% for clay).

**Comment # 81: Fig. S2-S24: please increase size of the letters in the legend. Present version is difficult to read.**

**Response:** Thank you for your suggestion. I have increased the size of the letters in the legend for Figures S2-S24 as recommended. This should enhance readability.

**Comment # 82: Fig. S2, S7, S8-13 : using the word "content" is not appropriate, please revise these captions. Fig. S8-13 needs some further clarification on the meaning of R, G, B, should be easy to understand without reading the manuscript.**

Response: Thank you for your valuable feedback. I have removed the term "content" from the captions of Fig. S2, S7, and S8-13, as you suggested. Additionally, I have clarified the meaning of R, G, and B in the captions for Fig. S8-13 to ensure that they are easily understandable without needing to refer to the manuscript. For example, in Fig. S8, the title now states:

"**Figure S8. The predicted maps of the red (R) component of soil color (Wet) at multiple depths. (a) 0-5 cm; (b) 5-15 cm; (c) 15-30 cm; (d) 30-60 cm; (e) 60-100 cm; and (f) 100-200 cm depth interval. The R component represents the red channel in the RGB soil color system.**"

**Comment # 83: Fig. S14-15: I thought there are more variety in the colour of the soil. Do you have only 6 different colour? Or did you decrease/aggregate the possible colours?**

Response: Thank you for your comment. You are correct that soil colours exhibit greater variety. In the visualization, we aggregated some less representative colours and displayed six key representative colours to improve clarity in the maps. This approach helps in effectively conveying the main patterns while maintaining visual simplicity.

The updated Figure caption now reads:

"**The Colour bar displays six representative colours, with some less distinctive colours aggregated for clarity.**"

**Comment # 84: Fig. S19-24: write out nitrogen, potassium, phosphorus before the brackets, instead of writing only N, K, and P.**

Response: Thank you for your suggestion. I have revised Fig. S19-24 to spell out nitrogen, potassium, and phosphorus before the brackets, as you recommended.

**Comment # 85: Fig. S2-25: please add unit in the caption of the figure.**

Response: I have added the units to the captions of Fig. S2-25 for the variables that have units. However, for maps such as the soil pH maps, which do not have units, the captions remain unchanged.

**Comment # 86: Fig. S25: ⋯ of the soil organic carbon (OC) and soil pH ⋯**

Response: Thank you for your feedback. We have modified the order of expression.

**Comment # 87: Fig. S26: please increase size of the letters on the plot, it is difficult to read.**

Response: I have increased the size of the letters on the plot for Fig. S26 as recommended. This should enhance readability.

**Comment # 88: Table S4: do you mean that it is the result of cross-validation in the case of CSDLv2 and performance of the other maps (CSDLv1, SoilGrids 2.0, HWSD 2.0) on the dataset used to train and test the CSDLv2 predictions? Please revise the title to increase clarity. Add number of samples considered for the validation in a separate column.**

**Response:** Thank you for your comment. Please refer to our response to comment #5. We have revised the title of Table S4 as per your suggestion to enhance clarity. Additionally, I have added a separate column to indicate the number of samples considered for validation.

**Comment # 89: Fig. S26: please add that the top 15 most important variables are shown.**

**Response:** Thank you for your comment. We have revised it as you suggested, and the revised Fig. S26 has the following caption:

**"Relative importance of the top 15 predictors for the Quantile Regression Forest model in the spatial predictions of soil total phosphorus (TP)......"**

---

## Author Comment (AC3)

Dear Reviewer,

Thank you very much for your comments and professional advice. Your insights have significantly contributed to enhancing the academic rigor of our article. We appreciate the time and effort you devoted to reviewing our work. Based on your valuable suggestions and requests, we have implemented corrections and modifications to the revised manuscript. We believe these enhancements will further strengthen the quality of our work. We would like to provide a detailed account of the changes made:

Note: The modifications are shown in bold font. The comments are blue colored.

**GENERAL COMMENTS:**

**Soil point data:**

**Comment #1: Key information is missing about the soil point data (Sect. 2.1.2). Are these observations (you use the term in-situ values) laboratory measurements or pedological field estimates (or perhaps both depending on the dataset and soil property)? If they are laboratory measurements, what methods were used to measure them? Are they data only from soil profiles or also from boreholes / augerings? At what depth was sampled (by fixed/predefined soil layer in cm or by pedological soil horizon)? What is the sampling design of the different datasets?**
**Response:** Thank you for your detailed comments, and I apologize for the confusion caused by the term "in-situ." The majority of the soil profile data used in this study are based on laboratory measurements rather than direct field observations ("in-situ"). We have revised the manuscript to replace "**in-situ**" with "**laboratory measurements**" to avoid any misunderstanding.

Regarding measurement methods, Shangguan et al. (2013) provide detailed descriptions for soil profiles from the Second National Soil Survey of China (SNSSC), while Batjes et al. (2020) document the measurement methods for soil profiles in the World Soil Information Service (WoSIS) database.

In response to your question on data sources, all observations are derived solely from soil profiles, with no data from boreholes or augerings. The regional database contains only surface data, and both the SNSSC and WoSIS datasets consist of soil profile data.

Concerning the sampling design, data collection was primarily soil type-based, with each soil type represented by one characteristic soil profile. Although the original soil surveys contained multiple profiles, only one representative profile was retained for each typical soil type in the final dataset. Since SNSSC soil profile data were extracted from soil survey books, there was no formal sampling design. However, if a sampling approach must be specified, it could be considered as a soil type-based stratified sampling design.

**Modification:**

"**The laboratory methods for soil profile data from the SNSSC and WoSIS databases are detailed in Shangguan et al. (2013) and Batjes et al. (2020), respectively. All data are exclusively from soil profiles, with no inclusion of boreholes or augerings. The regional database includes only surface data, while the SNSSC and WoSIS datasets contain full soil profiles. Sampling was primarily soil type-based, with each type represented by one characteristic profile. Although no formal sampling design was used for SNSSC data extracted from soil survey books, this approach may be considered soil type-based stratified sampling.**"

Shangguan, W., Dai, Y., Liu, B., Zhu, A., Duan, Q., Wu, L., Ji, D., Ye, A., Yuan, H., Zhang, Q., Chen, D., Chen, M., Chu, J., Dou, Y., Guo, J., Li, H., Li, J., Liang, L., Liang, X., Liu, H., Liu, S., Miao, C., and Zhang, Y.: A China data set of soil properties for land surface modeling, J. Adv. Model. Earth Syst., 5, 212–224, https://doi.org/10.1002/jame.20026, 2013.

Batjes, N. H., Ribeiro, E., and van Oostrum, A.: Standardised soil profile data to support global mapping and modelling (WoSIS snapshot 2019), Earth System Science Data, 12, 299–320, https://doi.org/10.5194/essd-12-299-2020, 2020.

**Data-splitting and model evaluation:**

**Comment #2: It seems that the authors did not group the data-splitting procedures by location / soil profile. If observations from the same profile but at different depths are used in both training and testing (calibration and validation, or in case of CV, it's also called hold-in vs. hold-out), then accuracy statistics are overly optimistic. This seems problematic in several steps of the modelling framework: RFE using OOB, 10-fold CV during hyperparameter tuning and most importantly, during model evaluation used for reporting the accuracy metrics. Please adjust methods so that, in all steps, all observations from the same location / profile are either in the hold-in or hold-out.**

**Response:**

Thank you for your insightful comment. In our study, we developed separate models for each soil depth layer individually, meaning that there is no overlap of observations from the same profile across training and testing datasets. This setup ensures that no observations from the same profile at different depths are used simultaneously in both the training and testing stages. Depth was not treated as a covariate, and each depth layer was modeled independently. We have clarified this approach in the manuscript and added notes in relevant sections and figures to specify that models were developed separately for each layer, avoiding the issue described.

**Modification:**

"**Separate models were developed independently for each soil depth layer, ensuring no overlap of observations from the same profile across training and testing datasets, with depth not used as a covariate.**"

**Discussion on use at various spatial scales:**

**Comment #3: I miss a discussion and recommendations of when and when not to use these maps. You have generated national maps for China of 20 soil properties, which you can expect will be widely used for science, policy and society. Therefore, it is in your interest to make sure they are not used the wrong way. Resolution is not the same thing as accuracy. While it's great that the authors have created high-resolution products, this does not mean that they are accurate or should be recommended to use at the local level, e.g. farm or field scale. For local-scale policy and land use decisions, local models with more detailed soil surveys would most likely need to be made. However, surely on a national scale and perhaps also on a large regional scale (provincial level), these maps can be used (given that users also consider the uncertainty that you report, i.e. accuracy metrics and uncertainty maps). Please add a section on this topic in the discussion supported by relevant literature.**

**Response:** Thank you for your suggestion. We have added a discussion to clarify the recommended spatial scales for using these maps. The new section highlights that, while these maps are high-resolution, they are best suited for national or regional applications, with additional caution advised for local-scale use.
**Modification:**

**"These maps are suitable for broad-scale applications, such as national and provincial-level analyses. Although generated at a high resolution (90 m), they may not provide sufficient accuracy for farm- or field-scale applications, where locally calibrated models and detailed surveys are recommended. Users should consider the provided uncertainty metrics to assess suitability for specific applications (Helfenstein et al., 2024)."**

Helfenstein, A., Mulder, V. L., Teuling, K., Walvoort, D. J. J., Heuvelink, G. B. M., Wageningen, A., and Wageningen, R.: BIS-4D: mapping soil properties and their uncertainties at 25 m resolution in the Netherlands, 2024.

**Comment #4: Please proofread for English spelling and grammar carefully. Currently there are numerous spelling and grammatical errors, some of which (not all) I have listed in the "technical corrections" below. Figures should be improved and legends and axes labels are often not readable.**
**Response:** Thank you for your careful review and helpful comments. We have thoroughly proofread the manuscript to address spelling and grammatical issues, making corrections throughout the text. Additionally, we have enlarged the font size of legends and axis labels in the figures to improve readability.

**Assets (data and code):**

**Comment #5: I was not able to access or download the data (90m resolution prediction maps). I recommend changing the data repository site and choosing one recommended by ESSD (https://www.earth-system-science-data.net/submission.html#assets). The model code is not provided and so the manuscript and modelling results are not reproducible (repository only contains 2 small scripts). I was not able to open the IGSN link when clicking on it but it did work when I pasted it into the browser (https://doi.org/10.11888/Terre.tpdc.301235). The "data sets" and "IGSN" assets are the same so one can be deleted. The "interactive computing environment" asset is merely a link to the python website and can be removed.**

**Response:** Thank you for your feedback. We have made the dataset available on an additional data platform, "scienceDB" (**https://www.scidb.cn/s/ZZJzAz**), which should facilitate smooth access to the 90 m resolution prediction maps. The repository now includes the model code for reproducibility. Currently, as the manuscript is in submission with ESSD, we are unable to remove the "IGSN" and "Interactive computing environment" links. However, if granted the permission to make changes later, we will remove these redundant assets.

**Modification:**

    **"The soil maps in this study for six depth layers (0-5, 5-15, 15-30, 30-60, 60-100, and 100-200 cm) at 90 m spatial resolution across China are openly accessible https://www.scidb.cn/s/ZZJzAz or https://doi.org/10.11888/Terre.tpdc.301235"**

*Specific comments:*

**Comment #6: L42-46: A more recent national product very similar to your own that is worth listing here is https://doi.org/10.5194/essd-16-2941-2024**

**Response:** Thank you for your suggestion. We have now included the recent national product by Helfenstein et al. (2024) on the Netherlands in the manuscript.

**Comment #7: L105-106: I would suggest to remove the first aspect: you already mentioned several times that new datasets were incorporated and more data were used than in other DSM studies in China. In addition, given the size of the country, the number of soil profiles is still not very high.**

**Response:** Thank you for your suggestion. We have removed the first aspect as recommended. We also acknowledge that, given the size of the country, the number of soil profiles remains limited.

**Modification:**

    **"Additionally, compared to existing datasets, this second edition offers a major innovation: over 20 comprehensive soil property variables were developed, while most current research focuses on mapping only a few basic soil properties."**

**Comment #8: L109-115: Thank you for including Fig. 2, which is very useful (see also my technical recommendations regarding this figure below). However, I think the list 1-4 here in the text does not summarize all the relevant steps completely. What about soil point data and covariate harmonization and preparation (which generally takes the longest!), model evaluation not only using data-splitting but also uncertainty maps?.**

**Response:** Thank you for your helpful comments. Based on your suggestion, we have revised the workflow to include the steps related to soil point data and covariate harmonization and preparation, as well as model evaluation using both data-splitting and uncertainty maps. These steps are now explicitly addressed in the updated workflow description.

**Modification:**

"The workflow of this study is shown in Fig. 1. Five main processes are involved in this framework:
1. **Harmonizing and preparing soil point data and environmental covariates.**
2. **Incorporating laboratory measurements of multiple soil profiles and overlaying them with covariates to generate a regression matrix for modeling.**
3. **Using cross-validation to obtain optimal modeling parameters.**
4. **Fitting prediction models based on the regression matrix.**
5. **Applying spatial prediction models using high-resolution covariates and evaluating the models using data-splitting and independent sample validation, as well as uncertainty maps."**

**Comment #9: L263-265: How did you obtain the mean prediction using QRF? Or did you use RF for obtaining the mean prediction? This issue is discussed also in https://doi.org/10.1016/j.geoderma.2021.115659IF: 5.6 Q1 , https://doi.org/10.5194/essd-16-2941-2024IF: 11.2 Q1 or https://doi.org/10.5194/soil-7-217-2021**

**Response:** Thank you for your insightful comment, and I apologize for the lack of clarity in the manuscript. In this study, we used the Random Forest (RF) model to obtain mean predictions. Quantile Regression Forests (QRF) were used specifically for generating prediction maps at different quantiles. We have clarified this distinction in the revised manuscript.

**Modification:**

"The RF model was used to generate mean predictions, while QRF were applied to produce prediction maps at different quantiles, providing a more comprehensive representation of uncertainty."

**Comment #10: L261-265: Did you compare median and mean predictions? You could do so quantitatively by comparing accuracy metrics and qualitatively by**

**comparing the quality of the maps visually. Perhaps for some of the many soil properties that you predicted, median predictions are more accurate or are to be preferred over mean predictions. Median and mean predictions of DSM products using QRF and RF are e.g. compared in https://doi.org/10.5194/essd-16-2941-2024.**

**Response:** Thank you for your valuable suggestion. We will conduct a comparison between the median and mean predictions to determine the most accurate approach for each soil property. This analysis may take some time, as certain properties may indeed perform better with median predictions, while others may be better represented by mean predictions. *Once this comparison is complete, we will update the final dataset based on the results as soon as possible.*

**Comment #11: L274: Why did the authors choose the WoSIS dataset as the independent dataset for statistical validation (second method)? Looking at Fig. 1 of the soil point data on the map, it's quite clear to me and it's a good choice, but it should still be shortly explained as this is an important detail. The choice of dataset used for statistical validation strongly influences accuracy metrics (e.g. https://doi.org/10.1111/j.1365-2389.2011.01364.x).**

**Response:** Thank you for your comment. As shown in the soil profiles spatial distribution map (Fig. 2), the WoSIS dataset has a more uniform spatial distribution across the study area, making it well-suited as an independent dataset for statistical validation.

**Comment #12: L276-281 and Eq. 2-4: Consider changing the order to ME followed by RMSE and then MEC since mathematically this makes much more sense (ME is a part of RMSE equation). This would also make more sense for explaining the terms in the text directly afterwards (L282-286).**

**Response:** Thank you for your suggestion. We have revised the order of ME, RMSE, and MEC as recommended,

**Comment #13: L303: I don't think Yan et al., 2020 is the most appropriate citation here. Better choose a manuscript that is specifically about prediction uncertainty and its error sources in DSM or statistical modelling. Some examples include: https://doi.org/10.1007/978-3-319-63439-5_14 or https://doi.org/10.1016/j.geoderma.2024.117052.**

**Response:** Thank you for your suggestion. We have updated the citation to refer to studies specifically addressing prediction uncertainty and error sources in DSM and statistical modeling, as recommended.

**Comment #14: L333-334: I suggest referencing the extensive review study of Chen et al. 2022 (https://doi.org/10.1016/j.geoderma.2021.115567) and also comparing with other studies (e.g. https://doi.org/10.5194/essd-16-2941-2024) not only in China to support the statement that pH is usually easiest to predict.**

**Response:** Thank you for your suggestion. We have updated the manuscript to include the recommended references, which provide further support for the statement that pH is usually the easiest soil property to predict, not only in China but also in other regions.

**Comment #15: L401: Careful! Confidence intervals are not the same as prediction intervals. Here you should be referring to prediction intervals, just as you do in the methods section.**

**Response:** Thank you for pointing this out, and I apologize for the oversight. We have revised the manuscript to replace "**confidence interval**" with "**prediction interval**" as suggested.

**Comment #16: L555-556: A more recent approach has also used covariates dynamic not only in two dimensional space but also over depth (and time), see https://doi.org/10.1038/s43247-024-01293-y.**

**Response:** Thank you for your suggestion. We have now included the recommended reference in the manuscript.

**Technical corrections:**

**Comment #17: L75: remove parentheses around Zhou et al., 2019a.**

**Response:** The parentheses around Zhou et al., 2019a have been removed as suggested.

**Comment #18: L104:** "**without explicit uncertainty**"

**Response:** Thank you for your comment. In response to your feedback and the suggestions from the first anonymous reviewer, we have made the following revisions: The key advancements of this second edition dataset, compared to the first edition, are as follows:

**1. Integration of multi-source soil profile samples, including data from the Second National Soil Survey of China (Shangguan et al., 2013), the World Soil Information Service (Batjes et al., 2020), the First National Soil Survey of China (National Soil Survey Office, 1964), and regional databases (Shangguan et al., 2012), enhancing the spatial representation of soil profiles, rather than relying solely on data from the Second National Soil Survey as in CSDLv1.**

**2. Application of advanced machine learning methods, replacing the conventional soil polygon linkage method used in CSDLv1.**

**3. Consideration of high-resolution environmental covariates as predictors for the machine learning models, allowing the model to capture more detailed spatial relationships between soil properties and environmental factors.**

4. **As a result of the improvements in points 1-3, the spatial resolution has been enhanced from the original 1 km to 90 m, providing more detailed and accurate spatial predictions of soil properties.**

**Comment #19: L109: Fig. 1 is the map of soil profiles. Here I assume you refer to Fig 2. Check this and make sure all tables and figures are in the correct chronological order in which they appear in the text.**

**Response:** Thank you for pointing this out. We have adjusted the order of the figures to ensure they appear in the correct sequence as referenced in the text.

**Comment #20: L131: If I am not mistaken 11,209 should be written as 11 209 and 8,979 as 8979. Also, there should be spaces between units (also percentages) and the number. Please carefully read through https://www.earth-system-science-data.net/submission.html. There is a very detailed and useful section about "mathematical notation and terminology". Please check this and apply to entire manuscript.**

**Response:** Thank you for your suggestion. We have revised "**11,209**" to "**11 209**" and "**8,979**" to "**8 979**" as recommended. Additionally, we have reviewed the entire manuscript and made the necessary formatting adjustments for all numerical values and units according to the guidelines.

**Comment #21: L146-147: include reference to GSM standard depths to make it clear which international standards you are referring to:**
**Arrouays et al., 2014. GlobalSoilMap: Basis of the global spatial soil information system)**
**Arrouays et al., 2015. The GlobalSoilMap project specifications, in: Proceedings of the 1st GlobalSoilMap Conference.**

**Response:** Thank you for your suggestion. We have added the recommended references (**Arrouays et al., 2014; Arrouays et al., 2015**) to clarify the international standards for soil depth used in this study.

**Comment #22: L160: Perhaps adjust to "Covariates related to the soil-forming factor 'organism'".**

**Response:** Thank you for your suggestion. We have revised the text to "**Organism-related covariates were primarily sourced from six datasets…**" as recommended.

**Tables and Figures:**

**Comment #23: Figures in the manuscript and supplements are often too small, axis and legend labels are non-readable. It is key that these figures are improved for publication, as maps are key to this study. Some colors scales in the figures are not color-blind friendly (red and green colors), e.g. Fig. S26.**

**Response:** Thank you for your valuable feedback. We have made improvements to the figures in both the main manuscript and the supplements to enhance readability and ensure they are suitable for publication. The axis and legend labels have been enlarged, and we have adjusted the color schemes to be more color-blind friendly, avoiding red and green combinations as suggested.

**Comment #24: In general, I would recommend re-assessing where and how information is presented in figures, which I realize is challenging with so many predicted soil properties at different depths and maps of uncertainty etc. Perhaps see https://doi.org/10.5194/essd-16-2941-2024IF: 11.2 Q1 and the supplements of that manuscript for ideas (https://doi.org/10.5194/essd-16-2941-2024-supplement) – there they organized the supplements by soil property.**

**Response:** Thank you for your suggestion. Following your recommendation, we have reorganized the figures and supplemental materials by soil property, referencing the structure provided in the ESSD manuscript (https://doi.org/10.5194/essd-16-2941-2024) and its supplements. We hope this improves the clarity and accessibility of the presented information.

**Comment #25: Figure 2: remove "altitude", shown in parentheses below depth. Altitude usually refers to elevation, whereas here you are referring to depth. According to Meinshausen 2006, QRF should be "quantile regression forest", not "quantile random forest". You also refer to it as quantile regression forest elsewhere. Check entire manuscript to make sure it's the same. "Variables" is misspelled ("varibles maps"). Finally, the caption is grammatically incorrect: either "for national-scale soil properties mapping" or "for developing national-scale soil property maps". Please check.**

**Response:** Thank you for your detailed feedback. We have removed "altitude" as suggested and corrected "QRF" to consistently refer to "**quantile regression forest**" throughout the manuscript. We have also fixed the spelling of "**Variables maps**" and revised the figure caption to "f**or developing national-scale soil property maps**" for grammatical accuracy.

**Comment #26: Figure 5: Maps are too small. Legends and axis labels cannot be read. Maps need to be enlarged. Consider restructuring figures (see comment above).**

**Response:** Thank you for your comment. We have enlarged the maps and increased the font size of legends and axis labels in Figure 5 to improve readability and suitability for publication.

---

## Author Comment (AC4)

Dear Reviewer,

Following your suggestion, we have compared the performance of all soil properties of interest in this study by evaluating both the mean predictions using Random Forest (RF) and median predictions using Quantile Regression Forest (QRF). The table below presents a 10-fold cross-validation performance comparison for each method—mean prediction by RF and median prediction by QRF—under the 'All Data', 'High Values' and 'Low Values' conditions. Specifically, the 'All Data' condition evaluates performance across the entire training set, while 'High Values' and 'Low Values' conditions focus on prediction accuracy for the top 10% highest and bottom 10% lowest values, respectively.

The 'Prediction method' column documents the models constructed to generate the final nationwide 90-meter resolution predictions for each soil property. In selecting the models, we considered their performance in mean predictions and their ability to capture extreme values (both maximum and minimum). Additionally, we observed a consistent trend in model performance across different depth layers for each soil property (i.e., for all layers of a specific property, either RF's mean predictions or QRF's median predictions consistently outperformed the other, as shown in the figure below). Consequently, for each specific soil property, only one optimal prediction model was ultimately selected to develop the 90-meter resolution soil maps. Therefore, we have presented only the performance metrics for the 0-5 cm surface depth in the table to streamline the comparison.

Thank you once again for your insightful comments, which helped us improve the clarity of our work.

We have added the following to the manuscript:
**Modification:**

**Although the performance differences between mean predictions using RF and median predictions using QRF are minimal, their ability to capture extreme values was considered. In this study, we evaluated the performance of RF and QRF models by not only the overall statistical metrics but also their capacity to predict extreme values (i.e. both high and low values), to determine the most suitable model for generating national gridded soil maps of various soil properties at a 90-meter resolution. As shown in Table S7, soil properties such as soil pH, silt, clay, TP, Red (R) of wet soil color, Blue (B) of wet soil color, Red (R) of dry soil color, and Blue (B) of dry soil color were modeled using median predictions from QRF, as this approach better captured extreme values. Similarly, the study by Helfenstein et al., (2024) also assessed mean predictions by RF and median predictions by QRF, highlighting that for certain soil properties, median predictions are more appropriate than mean predictions. For most other soil properties in this study—such as sand, BD, OC, gravel, AN, TN, CEC, porosity, TK, AK, AP, Green (G) of wet soil color, and Green (G) of dry soil color—mean predictions from RF were used to generate the 90-meter resolution soil maps. The better model was consistent across different depths for**

**the same soil property; thus, Table S7 only presents the performance comparison of mean and median predictions for the surface layer (0-5 cm depth interval) and either the mean or the median is used for the mapping of a soil property for all depths.**

**When developing the 90-meter resolution soil maps in this study, either mean or median predictions were selected for storage efficiency. However, for lower-resolution maps provided at 1 km and 10 km, in addition to mean and median predictions, we also included prediction maps for the 0.05 and 0.95 quantiles. These additional maps are helpful for illustrating data uncertainty.**

Helfenstein, A., Mulder, V. L., Teuling, K., Walvoort, D. J. J., Heuvelink, G. B. M., Wageningen, A., and Wageningen, R.: BIS-4D: mapping soil properties and their uncertainties at 25 m resolution in the Netherlands, 2024.

The information added in the supplementary material is presented in the table below:

**Table S1. Comparison of predictive performance for mean predictions using random forest model and median predictions using quantile regression Forest model across different soil properties under 'All Data,' 'High Values,' and 'Low Values' conditions based on 10-fold cross-validation. The 'All Data' condition evaluates performance on the full training set, while 'High Values' and 'Low Values' assess prediction accuracy for extreme high and low values within the training set, respectively. The 'Prediction method' column documents the models constructed for generating final national-scale predictions at a 90-meter resolution for various soil properties.**

| property | Statistic Validation | All Data | | | High Values | | | Low Values | | | Prediction method |
|---|---|---|---|---|---|---|---|---|---|---|---|
| | | MEC | RMSE | ME | MEC | RMSE | ME | MEC | RMSE | ME | |
| pH | Mean | **0.693** | **0.706** | **0.001** | -2.923 | 0.786 | -0.564 | -7.196 | 1.023 | 0.830 | **Median** |
| | Median | 0.690 | 0.709 | -0.012 | **-2.871** | **0.781** | **-0.557** | **-5.958** | **0.943** | **0.730** | **(QRF)** |
| sand | Mean | **0.670** | **12.161** | **0.056** | **-8.000** | **22.178** | **-16.260** | -9.543 | 12.961 | 8.400 | Mean |
| | Median | 0.667 | 12.231 | -0.734 | -8.612 | 22.919 | -16.560 | **-9.070** | **12.021** | **7.299** | (RF) |
| silt | Mean | 0.615 | 9.825 | 0.023 | -4.324 | 15.014 | -10.967 | -8.659 | 15.652 | 11.526 | **Median** |
| | Median | 0.614 | 9.840 | 0.003 | **-4.240** | **14.895** | **-10.789** | **-8.838** | **15.796** | **11.139** | **(QRF)** |
| clay | Mean | **0.629** | **6.749** | **0.019** | -1.328 | 12.197 | -8.221 | -23.577 | 8.543 | 6.279 | **Median** |
| | Median | 0.626 | 6.771 | 0.018 | **-1.281** | **12.071** | **-7.919** | **-23.416** | **8.515** | **6.088** | **(QRF)** |
| BD | Mean | **0.623** | **0.119** | **0.001** | **-2.230** | **0.188** | **-0.129** | **-0.561** | **0.208** | **0.133** | Mean |
| | Median | 0.619 | 0.120 | -0.000 | -2.351 | 0.192 | -0.133 | -0.619 | 0.212 | 0.140 | (RF) |
| OC | Mean | **0.570** | **2.043** | **0.028** | **0.089** | **5.382** | **-2.647** | -98.229 | 1.056 | 0.559 | Mean |
| | Median | 0.556 | 2.075 | -0.225 | -0.071 | 5.836 | -3.455 | **-69.297** | **0.889** | **0.464** | (RF) |
| gravel | Mean | **0.494** | **13.010** | **0.066** | **-5.133** | **24.486** | **-19.554** | -150.920 | 10.572 | 8.463 | Mean |

| | | | | | | | | | | |
|---|---|---|---|---|---|---|---|---|---|---|
| | Median | 0.483 | 13.152 | -1.542 | -5.427 | 25.067 | -19.771 | **-103.985** | **8.789** | **6.430** | (RF) |
| AN | Mean | **0.535** | **96.580** | **1.489** | **-0.671** | **224.419** | **-155.231** | -91.610 | 89.083 | 58.083 | Mean |
| | Median | 0.528 | 97.276 | -8.873 | -0.882 | 238.166 | -171.510 | **-80.097** | **83.362** | **51.067** | (RF) |
| TN | Mean | **0.437** | **0.153** | **0.003** | **-0.602** | **0.403** | **-0.249** | -63.525 | 0.090 | 0.066 | Mean |
| | Median | 0.411 | 0.157 | -0.024 | -0.950 | 0.445 | -0.310 | **-37.921** | **0.069** | **0.050** | (RF) |
| CEC | Mean | **0.342** | **8.516** | **0.168** | **-1.280** | **20.586** | **-15.277** | -47.768 | 7.887 | 6.714 | Mean |
| | Median | 0.322 | 8.644 | -1.273 | -1.706 | 22.427 | -17.579 | **-31.976** | **6.486** | **5.280** | (RF) |
| porosity | Mean | **0.286** | **5.496** | **-0.028** | **-6.014** | **9.548** | **-8.380** | **-10.608** | **10.167** | **9.167** | Mean |
| | Median | 0.283 | 5.507 | 0.064 | -6.041 | 9.566 | -8.436 | -10.728 | 10.219 | 9.236 | (RF) |
| TK | Mean | **0.254** | **0.569** | **0.004** | **-6.439** | **1.133** | **-0.985** | -7.496 | -0.921 | 0.842 | Mean |
| | Median | 0.251 | 0.570 | -0.022 | -6.856 | 1.164 | -1.007 | **-6.626** | **0.873** | **0.772** | (RF) |
| TP | Mean | 0.039 | 0.153 | **0.001** | **-0.073** | **0.471** | **-0.114** | -45.798 | 0.047 | 0.040 | **Median** |
| | Median | **0.042** | **0.153** | -0.012 | -0.092 | 0.475 | -0.136 | **-23.025** | **0.034** | **0.029** | **(QRF)** |
| AK | Mean | **0.161** | **169.589** | **1.120** | **-0.250** | **484.127** | **-235.202** | -74.502 | 91.844 | 77.809 | Mean |
| | Median | 0.130 | 172.666 | -24.174 | -0.413 | 514.801 | -285.971 | **-46.213** | **72.628** | **61.121** | (RF) |
| AP | Mean | **0.137** | **10.600** | **0.284** | **-1.000** | **29.102** | **-21.562** | -217.302 | 6.999 | 6.334 | Mean |
| | Median | 0.075 | 10.976 | -2.468 | -1.470 | 32.340 | -25.594 | **-90.100** | **4.521** | **4.074** | (RF) |
| R (Wet) | Mean | **0.275** | **33.108** | **0.032** | -10.615 | 56.055 | -50.741 | -10.427 | 54.593 | 49.311 | **Median** |
| | Median | 0.271 | 33.212 | 0.081 | **-10.481** | **55.730** | **-50.198** | **-10.363** | **54.441** | **48.539** | **(QRF)** |
| G (Wet) | Mean | **0.258** | **32.333** | **0.076** | **-12.180** | **55.557** | **-51.001** | -24.998 | 52.446 | 48.498 | Mean |
| | Median | 0.244 | 32.639 | -0.777 | -12.543 | 56.317 | -51.137 | **-24.089** | **45.522** | **46.730** | (RF) |
| B (Wet) | Mean | **0.205** | **34.046** | **0.021** | -9.174 | 57.428 | -52.629 | -75.942 | 54.758 | 50.755 | **Median** |
| | Median | 0.193 | 34.305 | 0.934 | **-8.686** | **56.034** | **-50.974** | **-74.629** | **54.168** | **49.383** | **(QRF)** |

| | | | | | | | | | | | |
|---|---|---|---|---|---|---|---|---|---|---|---|
| R (Dry) | Mean | **0.256** | **34.204** | **0.041** | -11.524 | 58.243 | -51.861 | -11.524 | 56.236 | 50.954 | **Median** |
| | Median | 0.249 | 34.331 | 0.095 | **-11.142** | **57.531** | **-51.256** | **-11.321** | **56.112** | **50.364** | **(QRF)** |
| G (Dry) | Mean | **0.269** | **31.238** | **0.067** | **-11.173** | **54.248** | **-50.843** | -23.128 | 50.571 | 46.368 | Mean |
| | Median | 0.254 | 31.854 | 0.421 | -11.534 | 55.658 | -50.994 | **-22.451** | **46.358** | **43.589** | (RF) |
| B (Dry) | Mean | **0.213** | **33.224** | **0.020** | -9.854 | 56.552 | -52.223 | -74.642 | 53.775 | 49.228 | **Median** |
| | Median | 0.204 | 33.612 | 0.635 | **-9.347** | **55.012** | **-50.128** | **-73.734** | **53.127** | **48.581** | **(QRF)** |

[Figure]

**Continue**

[Figure]

**Figure. Predicted median (a, c, e, g, i, k) and mean (b, d, f, h, j, l) bulk density (BD) at various depth on the y-axis vs. measured BD content on the x-axis. Accuracy plots and metrics (ME, RMSE and MEC) were computed using 10-fold cross-validation.**